Analysis

# An updated evolutionary classification of CRISPR–Cas systems including rare variants

Kira S. Makarova [1,38], Sergey A. Shmakov[1,2,38], Yuri I. Wolf [1], Pascal Mutz [1], Han Altae-Tran[3], Chase L. Beisel [4,5], Stan J. J. Brouns[6], Emmanuelle Charpentier [7], David Cheng[8], Jennifer Doudna[9,10,11,12,13,14], Daniel H. Haft[15], Philippe Horvath [16], Sylvain Moineau [17], Francisco J. M. Mojica [18], Patrick Pausch [19], Rafael Pinilla-Redondo [20], Shiraz A. Shah [21], Virginijus Siksnys [22], Michael P. Terns [23], Jesse Tordoff[8], Česlovas Venclovas [22], Malcolm F. White [24], Alexander F. Yakunin [25,26], Feng Zhang [27,28,29,30,31,32,33], Roger A. Garrett [34], Rolf Backofen [35], John van der Oost [36], Rodolphe Barrangou [37] & Eugene V. Koonin [1] ✉

The known diversity of CRISPR–Cas systems continues to expand. To encompass new discoveries, here we present an updated evolutionary classification of CRISPR–Cas systems. The updated CRISPR–Cas classification includes 2 classes, 7 types and 46 subtypes, compared with the 6 types and 33 subtypes in our previous survey 5 years ago. In addition, a classification of the cyclic oligoadenylate-dependent signalling pathway in type III systems is presented. We also discuss recently characterized alternative CRISPR–Cas functionalities, notably, type IV variants that cleave the target DNA and type V variants that inhibit the target replication without cleavage. Analysis of the abundance of CRISPR–Cas variants in genomes and metagenomes shows that the previously defined systems are relatively common, whereas the more recently characterized variants are comparatively rare. These low abundance variants comprise the long tail of the CRISPR–Cas distribution in prokaryotes and their viruses, and remain to be characterized experimentally.

Clustered regularly interspaced short palindromic repeats (CRISPR)–CRISPR-associated protein (Cas) systems are best known as the new generation of genome engineering tools[1,2]. However, their primary natural role is adaptive immunity in bacteria and archaea that functions by recognizing and, typically, cleaving a specific sequence in the target DNA or RNA that is complementary to a unique spacer between CRISPR repeats. The CRISPR–Cas immune response includes three main stages: adaptation, expression and interference, which are discussed in many dedicated reviews covering various aspects of the molecular mechanisms of CRISPR–Cas functionality[3–7]. CRISPR–Cas systems have a characteristic modular organization that roughly corresponds to the three stages of adaptive immunity (Fig. 1; details in Supplementary Note).

As expected of defence mechanisms, CRISPR–Cas systems show extensive diversity of the organization of their respective genome loci, *cas* gene composition as well as domain architectures and sequences of Cas proteins[3,4,8,9]. In the nearly two decades since the discovery of the CRISPR–Cas function[10], knowledge and understanding of this remarkable system has been steadily expanding through the mining of rapidly growing genomic and metagenomic databases. Classification of CRISPR–Cas systems based, to the maximum extent possible, on evolutionary relationships is essential for accurate description and further characterization of CRISPR-*cas* loci in newly sequenced bacterial and archaeal genomes and metagenomes, and hence, for the progress of the entire field of CRISPR research. However, CRISPR–Cas systems share no universal markers suitable for comprehensive phylogenetic

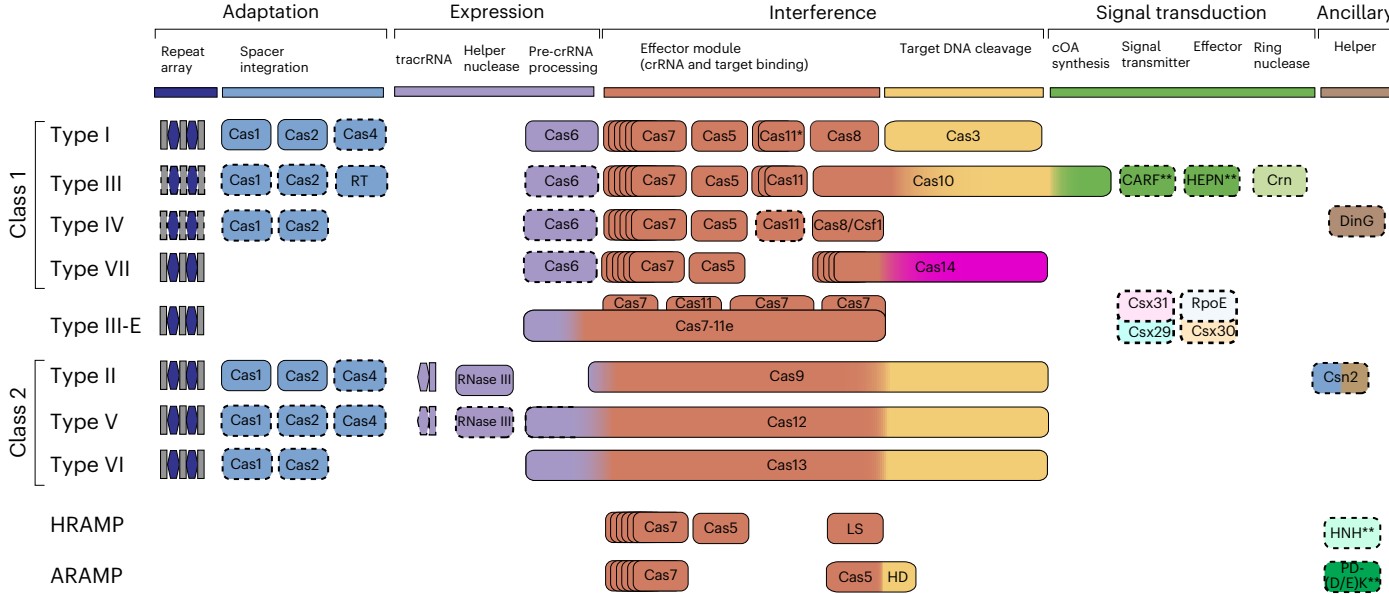

**Fig. 1 | Modular organization of CRISPR–Cas systems.** In class 1 CRISPR–Cas systems, effector modules consist of multiple Cas proteins that form a crRNA-binding complex and function together in target binding and cleavage. Class 2 systems have a single multidomain crRNA-binding protein that is functionally analogous to the entire effector complex of class 1. Subtype III-E is an exception within class 1, with a single effector protein composed of several domains derived from type III-D systems. The schematic shows the typical relationships between genetic, structural and functional organization for the seven types of CRISPR–Cas systems. Protein names follow the current nomenclature. Dispensable (and/or missing in some subtypes and variants) components are indicated by dashed outlines. Cas6 is shown with a thin solid outline for type I because it is dispensable in some but essential in most systems and with a dashed line for type III because most of these apparently use the Cas6 protein provided *in trans* by other CRISPR–*cas* loci. New type VII has a unique effector protein (Cas14) composed of two

domains: β-CASP family RNase fused to a domain homologous to the C terminal of Cas10. The three colours for Cas9, Cas10, Cas12 and Cas13 each reflect the fact that these proteins contribute to different stages of the CRISPR–Cas activity. The CARF and HEPN domains often fused in a single protein are the most common sensors and effectors, respectively, in the type III ancillary modules but several alternative sensors and effectors have been identified as well and are discussed in more detail in the main text. RING nucleases (Crn) cleave cyclic oligoA produced by Cas10 and thus control the indiscriminate RNase activity of the HEPN domain of a CARF protein[30]. HRAMP and ARAMP are array-less CRISPR-like systems derived from type III, which are typically associated with different nucleases, most often of HNH and PD-(D/E)xK families. *Putative small subunit that is fused to the large subunit in several type I subtypes. **This function can be performed by unrelated proteins; HD, HNH and PD-(D/E)K, nucleases of the respective superfamilies; LS, large subunit; RT, reverse transcriptase.

analyses, and both the organization of CRISPR–*cas* loci and the Cas proteins themselves evolve fast, which makes the construct of a consistent robust classification a major challenge. Three previous versions of CRISPR–Cas classification published in 2011, 2015 and 2020 (refs. 8,11,12) employed a complex polythetic approach. For the purpose of classification, comparisons the architecture and gene composition of CRISPR–*cas* loci were combined with clustering by sequence similarity and phylogenetic analysis of conserved Cas proteins, such as Cas1, the integrase that plays the key role in the adaptation stage. CRISPR–Cas types were delineated based on unique effector modules, whereas subtypes were defined less formally, based on a combination of the above criteria. A complementary development has been reported recently: Cas Protein Effector Database of Information and Assessment (CasPEDIA), a comprehensive classification of class 2 Cas enzymes based on their activity and target specificity[13].

The 2020 CRISPR–Cas system classification included 6 types and 33 subtypes partitioned between two classes that differ in the architectures of their effector modules involved in CRISPR RNA (crRNA) processing and interference[8]. Since then concerted efforts have been undertaken to identify additional CRISPR–Cas systems and to decipher their evolutionary origins, resulting in the discovery of type VII, 13 additional subtypes and numerous unique variants (Extended Data Figs. 1–4). Compared with previously identified CRISPR–Cas systems, these recently discovered systems are rare, apparently coming from the long tail of the CRISPR–Cas distribution among prokaryotes that remains to be further explored. In addition, many previously unrecognized CRISPR-linked genes have been discovered and the mechanisms of several CRISPR–Cas subtypes have been elucidated.

In this Analysis we describe the latest updates to the CRISPR–Cas classification and discuss the prospects of further discoveries being made as well as the current understanding of the main routes of the evolution of CRISPR–Cas systems.

## Results

### Updated classification of CRISPR–Cas systems

**Distinct features in class 1 CRISPR–Cas systems.** Although numerous class 1 CRISPR–Cas systems have been discovered since the 2020 classification, only type VII and two subtypes (III-G and III-H) have been explicitly added to the classification[9]. While working on this classification update, we identified a distinct variety of type III CRISPR–Cas systems, which seems to qualify as subtype III-I, bringing the total number of subtypes in class 1 to 21 (eight in type I, nine in type III, three in type IV and one in type VII (Figs. 1 and 2a, Extended Data Figs. 1 and 2, and Supplementary Note).

The previously undescribed type VII is represented by CRISPR–Cas systems found mostly in several taxonomically diverse archaeal genomes and containing a metallo-β-lactamase (β-CASP) effector nuclease[9,14]. According to the CRISPR–Cas classification principles, the unique signature effector shared by these systems, designated Cas14, qualifies these loci as a new type. Cas14 is encoded in a predicted operon with Cas7 and Cas5, the subunits of the effector complex, and in some cases Cas6, which is a dedicated nuclease involved in crRNA processing in other class 1 systems (Fig. 2a). Type VII loci lack adaptation modules and repeats in the associated CRISPR array often contain multiple substitutions, suggesting that the arrays do not frequently incorporate new spacers. Analysis of the limited

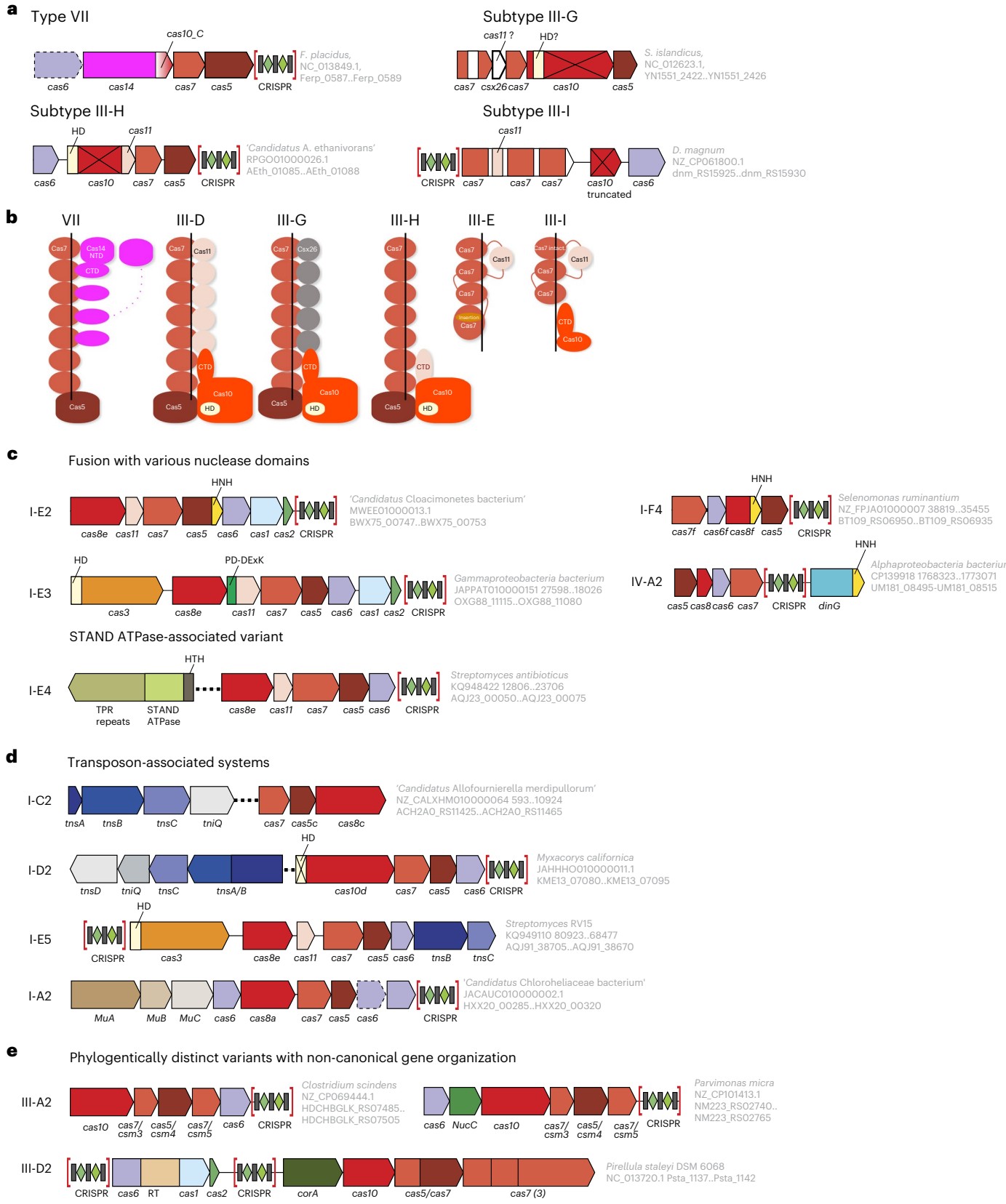

number of spacer hits indicates that these systems target transposable elements[9]. Notably, in addition to the β-CASP domain, the Cas14 protein contains a carboxy-terminal domain that structurally resembles the C-terminal domain of Cas10, the large subunit of the type III effector module, suggesting an evolutionary connection between types III and VII (Fig. 2a). This connection is also supported by the specific similarity between the Cas5 proteins of type VII and subtype III-D[9]. The catalytic residues of Cas7 that are required for the target RNA cleavage in type III systems are not conserved in type VII. Instead, type VII systems have been shown to target RNA

**Fig. 2 | New class 1 CRISPR–Cas systems. a**, Genetic organization of the recently identified type VII and subtypes III-G, III-H and III-I CRISPR–Cas systems. Homologous domains are colour-coded and identified by a family name. The gene names follow the previous classification[12]. Protein-coding genes are shown roughly to scale; CRISPR arrays are not shown to scale. In subtypes III-G, III-H and III-I, the Palm domains of Cas10 responsible for signalling-molecule synthesis are inactivated. The organism and the corresponding gene range are shown in grey on the right of the loci. If genes are not present in every locus of the respective systems, they are shown by dashed outlines. In the type III-G schematic, the question mark next to Csx26 indicates that the assignment of this protein as a Cas11 counterpart remains hypothetical. **b**, Schematic representation of CRISPR–Cas effector complexes of type VII and subtypes III-D, III-E, III-G, III-H and III-I. The type VII, subtype III-D1 and III-E schematics are based on the solved structures (Protein Data Bank (PDB) identifiers 8zwl, 8bww and 7y82, respectively). The schematics for subtypes III-G and III-H are based on AlphaFold3 complex models using the following proteins and RNAs: III-G (NC_012623.1, *Sulfolobus islandicus*

Y.N.15.51): 6×Cas7 (most 5′), 1×Cas7 (between *csx26* and *cas10*), 5×Csx26, 1×Cas5, 1×Cas10, crRNA (48 nucleotides (nt)); III-H (RPGO01000026 'Candidatus Argoarchaeum ethanivorans', AEth_01085): 7×Cas7, 1×Cas5, 1×Cas10, crRNA is the same as for III-G; III-I (*Desulfonema magnum* WP_207677910 and WP_207677911.1): 1×Cas7-11i, 1×Cas10, crRNA is from type III-E of *D. magnum* (PDB 7zol). AlphaFold3 interface predicted template modelling scores: III-G, 0.52; III-H, 0.59; III-I, 0.75. The black line schematically represents crRNA. Full models in Extended Data Fig. 5. **c**, Genomic locus organization of type I and type IV variants that acquired a new nuclease domain and/or lost Cas3 helicase-nuclease. Designations are as in **a**. **d**, Genomic locus organization of new Tn7 and Mu transposon-associated type I CRISPR–Cas variants. The *tns* and *tni* genes encode transposase subunits. Designations are as in **a**. **e**, Genomic locus organization of new type III variants. Designations are as in **a**. PD-(D/E)xK, nuclease of the respective superfamilies; TPR, tetratricopeptide repeats; STAND, signal transduction ATPases with numerous domains; NTD, N-terminal domain; CTD, C-terminal domain.

---

in a crRNA-dependent manner, cleaving the target via the nuclease activity of Cas14 (Supplementary Table 1). Type VII systems seem to be simple and have probably evolved from type III via the reductive route as discussed below. However, the recently solved cryogenic-electron-microscopy structure of the type VII effector complex contains up to 12 subunits, with Cas14 binding to the Cas7 backbone via its Cas10 remnant domain, making this complex one of the largest among the class 1 systems[14] (Fig. 2a and data file 1 in ref. 15) https://doi.org/10.5281/zenodo.15882620.

The three previously undescribed subtypes of type III CRISPR–Cas systems—that is, III-G (Sulfolobales-specific, previously reported as unclassified[8]), III-H (present in various archaea and a few bacterial metagenome-assembled genomes (MAGs)) and III-I (present in more than 160 genomes in the National Center for Biotechnology Information (NCBI) non-redundant (NR) database, mostly from the phyla Thermodesulfobacteriota and Chloroflexota)—are not closely related but share some features that suggest reductive evolution[9] (Fig. 2a). In subtypes III-G and III-H, the polymerase/cyclase domain of Cas10, the large subunit of the effector complex, is inactivated as indicated by the replacement of the catalytic amino acids. The lost capacity to generate cyclic oligoadenylates (cOAs) correlates with the loss of genes encoding ancillary proteins containing a cOA-binding domain (such as CRISPR-associated Rossmann fold (CARF) or SMODS (second messenger oligonucleotide or dinucleotide synthetase)-associated and fused to various effector domains (SAVED)) fused to an effector domain, such as higher eukaryotes and prokaryotes nucleotide-binding (HEPN) RNase or other effectors that are typical of type III CRISPR–Cas systems[8]. Thus, these subtypes have lost the cOA signalling pathway that induces collateral RNase activity in most type III systems. Although subtype III-H is distantly related to III-F, a unique feature of III-H systems is a highly diverged small subunit (Cas11) that apparently has replaced the C-terminal domain of Cas10 (Fig. 2a). In subtype III-G, Csx26, the signature protein of this subtype, might replace Cas11 in the effector complexes (Fig. 2b and Extended Data Fig. 5). Both these new subtypes lack adaptation modules and no CRISPR array has so far been found in III-G loci, suggesting that this system recruits crRNAs from other CRISPR–*cas* loci *in trans*. Given the lack of conservation of catalytic aspartates in Cas7 and the presence of apparently active HD-nuclease domains in Cas10, both III-G and III-H are predicted to cleave DNA targets (Fig. 2a, Extended Data Fig. 2a and Supplementary Table 1).

The effector module of the new subtype III-I systems (Fig. 2a,b, Extended Data Figs. 2 and 6 and data file 2 in ref. 15) discovered during the present analysis has two unique features: (1) an extremely diverged Cas10 lacking the amino-terminal polymerase/cyclase domain; this protein lacks detectable sequence similarity to Cas10 but was confidently identified as a Cas10 homologue in the structure similarity search (distance-matrix alignment (DALI) Z-score = 10.9);

and (2) a multidomain protein with a domain architecture resembling that of Cas7–11, the effector protein of subtype III-E, but apparently originating independently from a different variant of subtype III-D (Extended Data Fig. 6). The III-I effector protein consists of three fused Cas7 domains and a Cas11 domain that lacks both the N-terminal Cas7 domain and an insertion in the C-terminal Cas7 present in Cas7–11 (Extended Data Fig. 6). Based on the presence of conserved aspartates in each of the three Cas7 domains, the subtype III-I CRISPR–Cas system most probably cleaves RNA (data file 2 in ref. 15). We propose to denote the III-I effector Cas7-11i and accordingly amend the designation for the III-E effector to Cas7-11e.

In addition to type VII and the three previously undescribed subtypes, multiple variants of class 1 CRISPR–Cas systems with unique domain architectures and functional features have recently been discovered (Fig. 2c). Three of these variants (I-E2, I-F4 and IV-A2) encompass an HNH nuclease that is fused to Cas5, Cas8f and CasDinG proteins, respectively[9]. Robust crRNA-guided double-stranded DNA (dsDNA) cleavage activity has been demonstrated for each of these variants[9,16,17]. Notably, the I-E2 and I-F4 variants typically lack Cas3 helicase-nuclease, which is responsible for the shredding of the DNA target in most type I CRISPR–Cas systems, so that the HNH nuclease seems to replace the nuclease activity of the HD-nuclease domain of Cas3. The HNH nuclease is typically encoded by mobile genetic elements (MGEs) such as group I self-splicing introns[18]. Cas9, the type II effector, also contains an HNH nuclease that is inserted into the other RuvC-like nuclease domain of Cas9 and is responsible for the target DNA strand (the strand that hybridizes to the crRNA during R-loop formation) cleavage. The discovery of the new HNH-encoding variants shows that this mobile nuclease has been coopted by different CRISPR–Cas systems on multiple independent occasions. However, unlike the case of Cas9 where the HNH nuclease is responsible for the cleavage of only one DNA strand in the dsDNA target, in the new variants HNH is the only effector nuclease that cleaves both strands[9,16,17]. The IV-A2 variant is of special note, being the first type IV system shown to cleave the target. All other type IV-A and IV-B systems lack effector nucleases, apparently suppressing MGE reproduction via inhibition of transcription, at least in the case of IV-A[9,19]. Type IV-C systems are predicted to target and cleave DNA via the HD domain but remain experimentally uncharacterized. Another new variant, I-E3, encompasses a distinct nuclease of the PD-(D/E)xK family that is fused to Cas11[9]. Some of these systems contain Cas3, including an apparently active HD-nuclease domain, so the role of the PD-(D/E)xK nuclease remains to be determined.

Another recurrent trend in CRISPR–Cas evolution is the recruitment of CRISPR–Cas systems by large transposons, enabling RNA-guided transposition. The CRISPR-associated Tn7-like transposons (CASTs) of subtypes I-F and I-B were introduced in the previous classification[8] as I-F3 and I-B2 variants, respectively. More recently, three additional CAST varieties derived from subtypes I-D, I-C and

IV-A[20–22] have been discovered and a I-A system was found to be associated with Mu-like transposons[9] (Fig. 2d). Unlike most CASTs in which CRISPR–Cas effectors are inactivated (discussed later), some of the I-D CASTs are active, apparently representing the most recent capture of a CRISPR–Cas system by a transposon and, possibly, perform a dual function, acting both as a CAST and as a bona fide CRISPR–Cas system[21]. The I-A CASTs associated with Mu-like transposons have not yet been characterized experimentally and are expected to interact with the transposition machinery in a distinct manner because these transposons lack the TniQ–TnsD protein, an essential component of all previously discovered CASTs[21]. Finally, a distinct CAST variety (I-E5), also lacking TniQ–TnsD, was recently found in association with a distinct class of telomeric transposons[23]. This CAST contains an active HD-nuclease domain in the Cas3 protein and might also perform a dual function. All CASTs, except for subtype IV-A, of which so far only one instance has been found, were included in the current classification as variants of the respective CRISPR–Cas subtypes (Fig. 2d and Extended Data Figs. 1 and 2).

Other rare derivatives of class 1 systems continue to be discovered, for example, a type IV variant with a *cas* core gene set of apparent chimaeric origin, with Cas7 derived from IV-B and Cas5 from IV-A, and associated with a RecD-like helicase[24,25], or a I-F variant that seems to be an intermediate between I-F1 and I-F2 variants (Extended Data Fig. 1). Despite low abundance, characterization of such variants can potentially shed light on CRISPR–Cas evolution and reveal novel functionalities.

**Diversity of sensors and effectors in type III systems.** Most of the type III CRISPR–Cas systems are markedly more complex than other types including a striking variety of accessory proteins[26,27]. In the updated classification, type III is divided into nine subtypes, each of which encompasses a specific set of core effector genes and forms a distinct clade in the phylogenetic tree of Cas10 (Figs. 1 and 3a). The exceptions are subtype III-E, which lacks a *cas10* gene, and subtype III-I, which has an extremely diverged Cas10 derivative. Nevertheless, comparison of loci organization and phylogenetic analysis of Cas7 clearly shows that both III-E and III-I are derived from different variants of subtype III-D[28] (Extended Data Figs. 2a and 6).

The diversity of type III CRISPR–Cas systems is not adequately covered by the classification into subtypes and variants, requiring a more nuanced approach. A central feature of type III is the built-in signalling pathway in which, in response to target recognition, the polymerase/cyclase domain of Cas10 synthesizes either cOA or *S*-adenosyl methionine (SAM)–AMP, second messengers that are bound by the sensor domain of an ancillary protein. Binding of the messenger molecule induces a conformation change in the ancillary protein, resulting in activation of its effector domain (most often an HEPN RNase but in some variants distinct nucleases or other effectors), which causes growth arrest[29] (Fig. 3b). In addition, these signal transduction pathways are attenuated either by ring nucleases cleaving cOA[30,31] or by enzymes responsible for cleavage of SAM–ATP[32] (Fig. 3b). Sensors, effectors and ring nucleases in these pathways come in multiple forms resulting in combinatorial diversity[26]. Although the polymerases/cyclase domain of Cas10 is occasionally inactivated (subtypes III-C, III-G and III-H) or lost (subtype III-I), about 97% of the Cas10s are predicted to be active and capable of producing messengers (Fig. 3a, Extended Data Fig. 2 and data file 3 in ref. 15).

The cOA sensors in type III CRISPR–Cas systems are almost exclusively CARF and related SAVED domains[15,26,33]. Whereas many ring nucleases also possess CARF domains, several ring nucleases with enzymatic domains unrelated to CARF have also been identified[31]. Classification of CARF domains by sequence similarity led to the identification of 11 families, seven of which mostly consist of proteins containing both sensor (CARF) and effector (mostly, HEPN RNase but also other enzymes) domains, three include known or predicted ring nucleases and one (Csm6) consists of proteins apparently performing all three functions[31,33] (Fig. 3b, Supplementary Table 2 and data file 4 in ref. 15). Here we introduce a systematic nomenclature for the major groups of the sensor and ring nuclease CARF domain-containing proteins (Crf1–11 families; Crf is an abbreviation of CARF that we introduce to differentiate it from the generic domain name) to facilitate systematic analysis and description of type III CRISPR–Cas systems and the relationships among them. To ensure compatibility with the published work, legacy names are also indicated if they exist (Supplementary Table 2). All experimentally characterized families of ring nucleases are designated as Crn1–Crn5, with legacy names also retained (Supplementary Table 2).

The most common cOA-activated effectors are HEPN RNases but a variety of other unrelated effectors have been identified, including RNases of the PIN and RelE families, DNases of the PD-(D/E)xK and HD families, proteases of the Caspase and Lon families, several other enzymes as well as transcriptional regulators and integral membrane proteins[26] (Fig. 3b). Whereas the cOA pathway is well-established, components of the SAM–AMP pathway have not been thoroughly characterized. A membrane protein homologous to the magnesium transporter CorA that was identified in some III-B and III-D loci and shown to sense SAM–AMP via a dedicated sensor domain seems to be a signature of this pathway[32]. However, it remains unclear if the CorA homologue in III-D systems senses SAM–AMP or another signalling molecule, especially given that, in contrast to the III-B loci, known SAM–AMP cleavage enzymes are not encoded in the III-D loci (Fig. 3a,b).

Another notable clade in the Cas10 tree is the III-A2 variant, which lacks an identifiable Cas11 (Figs. 2e and 3a). In this variant, Cas10 is predicted to synthesize cOA, which in some of the systems probably directly activates the NucC nuclease effector[34]. The conservation of the catalytic aspartate in Cas7 suggests that this system might target both RNA, via Cas7, and DNA, via NucC (data file 3 in ref. 15).

Both sensors and effectors are scattered across the clades of the Cas10 tree, which is suggestive of extensive module shuffling and horizontal gene transfer shaping of the cOA and SAM–AMP signalling pathways in type III systems (Fig. 3a and data file 3 in ref. 15). Despite the extensive shuffling of signalling pathway components, some trends are notable[33,35]. First, ring nucleases are an important component of the signalling pathway and are present in most of the loci encoding enzymatically active Cas10 capable of cOA synthesis (Fig. 3a and data file 3 in ref. 15). Second, SAM–AMP is cleaved by a distinct set of enzymes typically encoded in the respective loci (Fig. 3a and Supplementary Table 2). Investigation of exceptions from these trends might lead to the discovery of Cas10s synthesizing different messengers as well as corresponding novel sensors and nucleases. A proof of principle is the identification of the Crf1_Csm6 family of proteins with a dual function of sensor and ring nuclease associated with III-A loci in which no other ring nuclease was detected[36]. Other cases such as the aforementioned CorA-associated III-D systems require further study.

**Fig. 3 | The built-in signalling pathway of type III CRISPR–Cas systems.**
**a**, Phylogenetic tree of Cas10 with information on associated genes involved in the signalling pathway and domain functionality mapped to the branches. Branches are colour-coded according to the key on the right. The circles around the tree show (from the inner to the outer): (1) inactive PALM (iPALM) domain, indicating that the respective Cas10 cannot make a second messenger molecule; (2) the presence of the HD-nuclease domain; (3)–(6) the presence of genes encoding Crf1, Crf2, Crf4 and Crf9 in the respective loci; (7) the presence of a gene encoding the CorA homologue; and (8) the presence of genes encoding SAM–AMP cleaving enzymes. **b**, Schematic of the signalling pathway. HEPN, PIN and RelE, ribonucleases of the respective families; CorA, divalent cation channel; HTH, helix-turn-helix domain. *In some Crf4 proteins, PD-(D/E)xK nuclease is capable of cleaving both DNA and RNA.

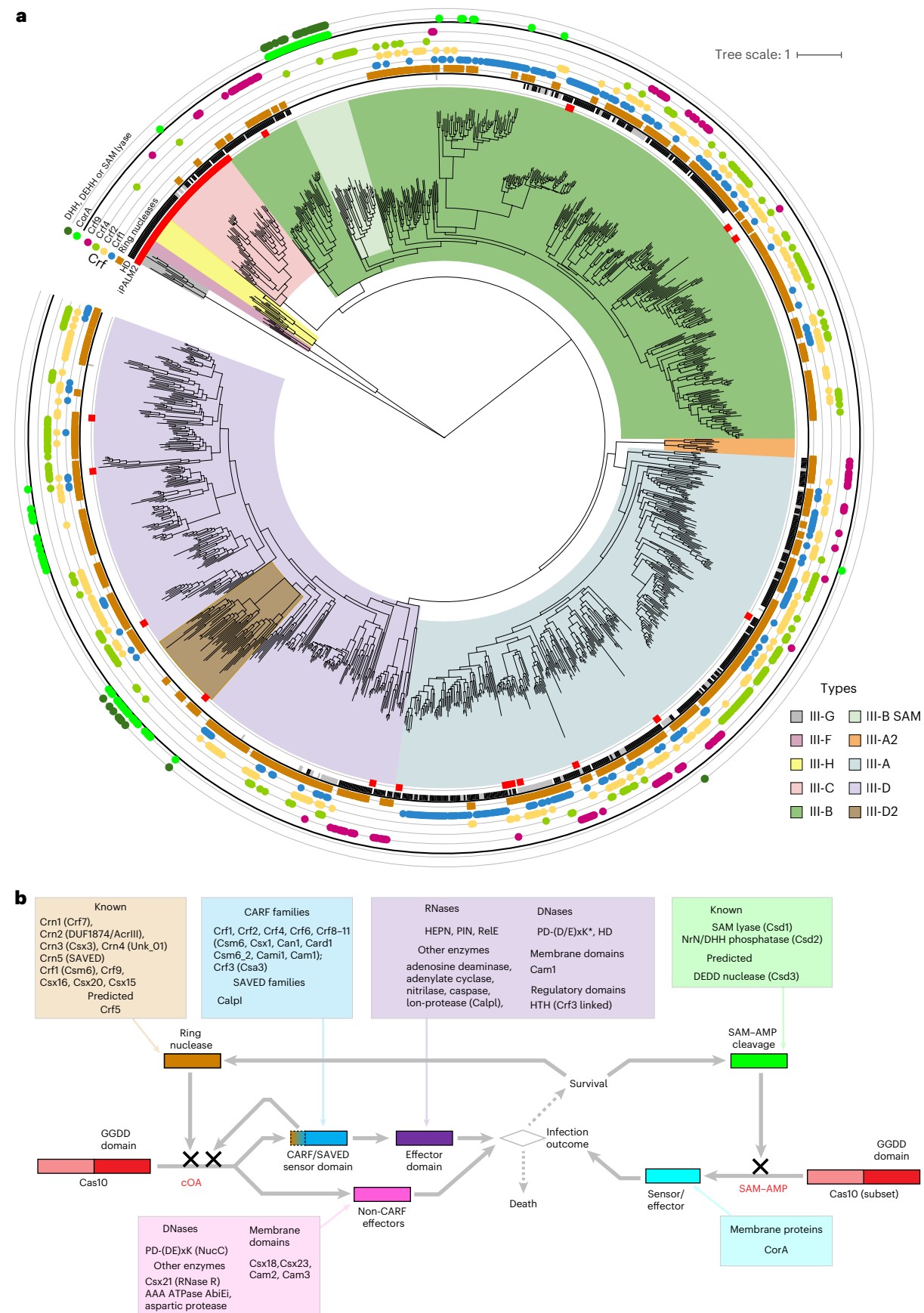

**Expansion of class 2 CRISPR–Cas systems.** The discovery of numerous class 2 CRISPR–Cas systems, primarily subtypes of type V, was the hallmark of the 2020 update of the CRISPR–Cas classification[8]. The distinguishing feature of these class 2 CRISPR–Cas systems is that their effector machinery consists of a single multidomain protein, namely Cas9 in type II, Cas12 in type V and Cas13 in type VI (Fig. 1 and Extended Data Figs. 3 and 4). Class 2 CRISPR–Cas systems are more simply organized than those of class 1, so classification relies primarily on comparison and phylogenetic analysis of the single large effector signature proteins. The continued discovery of distinct variants in the last few years led to the delineation of subtype II-D. This subtype includes the former II-C2 variant characterized by small archaeal Cas9s[8], designated II-D in CasPEDIA[13], together with the recently described variant encompassing the smallest known possibly ancestral Cas9s and also designated II-D[37]. Despite variation in size, phylogenetic analysis shows that Cas9d proteins confidently group together and form a deep branch in the Cas9 tree (Fig. 4a and data file 4 in ref. 15). Furthermore, the respective loci share the adaptation module containing *cas1*, *cas2* and *cas4* genes, with all Cas1 proteins forming a monophyletic branch in the Cas1 tree, supporting the robust classification of these loci as a single new subtype (Fig. 4a,b, Extended Data Fig. 3a and data file 4 in ref. 15). Notably, II-D systems are common among symbiotic and parasitic nanoarchaea of the Diapherotrites, Parvarchaeota, Aenigmarchaeota, Nanoarchaeota and Nanohaloarchaeota (DPANN) superphylum, along with some bacteria, whereas previously class 2 systems were considered a bacterial staple absent from the archaea. It has been shown that, like all type II CRISPR–Cas systems, type II-D loci encode trans-activating CRISPR RNA (tracrRNA)[37]. Given that, like all other archaea, DPANN members encoding type II-D CRISPR–Cas systems lack RNase III, the pre-crRNA in this case is likely to be processed either by an alternative RNase or by Cas9d itself. A rare, but notable, variant II-C2 encoded in phage genomes encompasses a Cas9 protein with an inactivated RuvC nuclease domain but apparently active HNH nuclease, which might function as an anti-CRISPR protein[9] (Fig. 4b).

As in the 2020 CRISPR–Cas classification[8], the greatest proliferation of subtypes and variants was observed within type V. Comprehensive phylogenomic analysis of transposon-encoded RNA-guided TnpB nucleases, the evolutionary ancestors of Cas12, showed that emergence of new type V CRISPR–Cas variants through the association of TnpB with CRISPR arrays and/or adaptation modules is not a rare event in the evolution of bacteria and archaea[38]. About 50 such independent events have been inferred and, beyond doubt, many more remain to be discovered in the ever-expanding genomic and metagenomic databases (Extended Data Fig. 4b and data files 5 and 6 in ref. 15). It is not always easy to decide which of these loci qualify as subtypes and which should remain variants. Variants can be upgraded to subtypes as they are studied experimentally and/or expanded through continued genome and metagenome exploration. In the last few years, six previously uncharacterized type V subtypes and 12 variants have been studied experimentally[9,39] (Fig. 4c–e and Extended Data Fig. 4a,b). At least five additional variants seem to be strong subtype candidates based on phylogenomic analysis (Extended Data Fig. 4b). One notable trend in type V evolution is the flexibility of target and collateral substrate recognition, including nucleases that can target and cleave both single-stranded (ss) RNA

and ssDNA, such as Cas12g[40], or are involved in both RNA targeting and collateral cleavage of ssRNA, ssDNA and dsDNA, such as Cas12a2 (ref. 41). Another trend is the inactivation of the RuvC-like nuclease domain of Cas12 that occurred independently on multiple occasions. At least some of these inactivated type V subtypes and variants nevertheless retain the interference capacity via mechanisms that do not include target DNA cleavage, such as blocking transcription via crRNA-guided binding of Cas12 to the target as demonstrated for subtypes V-C[42] and V-M[43,44], and confidently predicted for V-O and V-P, and possibly Cas12b3 (Extended Data Fig. 7). Other inactivated type V subtypes have been repurposed by transposons for RNA-guided transposition (discussed later; Extended Data Fig. 4b).

A signature feature of type II CRISPR–Cas systems is the involvement of tracrRNA in crRNA maturation and interference[45]. Among the type V systems, some also use tracrRNA (and are thus predicted to employ RNase III for pre-crRNA processing), whereas pre-crRNA processing is catalysed by a distinct active site in Cas12 itself in at least half of the characterized subtypes[40]. Thus, tracrRNA apparently evolved independently on many occasions in type II and type V CRISPR–Cas systems[46].

Two previously unrecognized subtypes of type VI, namely VI-E and VI-F, have been established. Both Cas13e and Cas13f are deep branches in the Cas13 dendrogram (Fig. 4e,f and Extended Data Fig. 3b–d). Cas13e is by far the smallest, most compact Cas13 protein and a potential early intermediate in type VI evolution[47]. Subtype VI-F, which was mentioned as an unclassified variant in the 2020 CRISPR–Cas classification is so far limited to a single bacterial genus, *Brachispira*. Structure analysis showed that Cas13b proteins share features distinct from those of other type VI effectors, suggesting the possibility of two independent origins of Cas13 from HEPN-containing toxins[47]. The group of subtype VI-B systems with the smallest known effectors are referred to as variant VI-B3 (Extended Data Fig. 3d).

Beyond the main effector nucleases, many class 2 CRISPR–Cas systems encompass a growing set of accessory proteins that augment the immune response, including some reported or characterized since the publication of the 2020 CRISPR–Cas classification. Subtype VI-B systems show a particularly notable diversity in both Cas13b size and the presence of associated genes[47,48]. Csx27 and Csx28 were previously identified as accessory proteins and shown to augment the immune function of Cas13b[49]. It has subsequently been shown that, following Cas13b activation, Csx28 forms an octameric pore that depolarizes the inner membrane[50]. Some of the other accessory proteins in class 2 CRISPR–Cas systems include the predicted PIN-domain nuclease PcrIIC1 that is encoded in some II-C loci and enhances the efficiency and promiscuity of DNA targeting[51,52] and an RNase H fold nuclease of the DUF3800 family encoded in some VI-F and V-A2 loci (Fig. 4g). More accessory proteins associated with class 2 systems and enhancing their immune function via different mechanisms probably await discovery.

**Distribution of CRISPR–Cas systems in bacteria and archaea.** As generally expected of antivirus defence systems[53], the different types and subtypes of CRISPR–Cas systems are highly non-uniformly distributed among bacterial and archaeal lineages at all levels, from phyla to isolates within species. We surveyed CRISPR–*cas* loci in the current collection of complete bacterial and archaeal genomes. Among

**Fig. 4 | New class 2 CRISPR–Cas systems. a,** Schematic of the relationships between type II subtypes based on Cas9 phylogenetic analysis (Extended Data Fig. 6). New subtypes and variants are shown in blue. **b,** Loci organization of new type II subtypes and variants. **c,** Schematic of the relationships among new and previously experimentally characterized type V subtypes based on the UPGMA dendrogram (Extended Data Fig. 4b) and information on their function and organization. Y, presence of the feature; p, partial presence of the feature; n, absence of the feature. 'Collateral' refers to indiscriminate cleavage of single-

stranded RNA or DNA by Cas nucleases. **d,** Dendrogram for new experimentally characterized subtype V-F variants (Extended Data Fig. 4b) and information on their functionality and organization. **e,** Loci organization of the new experimentally characterized type V subtypes and variants. **f,** Schematic of the relationships between type VI subtypes (Extended Data Fig. 3c). New subtypes and variants are shown in blue. **g,** Loci organizations of new type VI subtypes and variants. In **b**, **e** and **g**, the organism and the corresponding gene range are shown in grey on the right of the loci. Inact, inactivated.

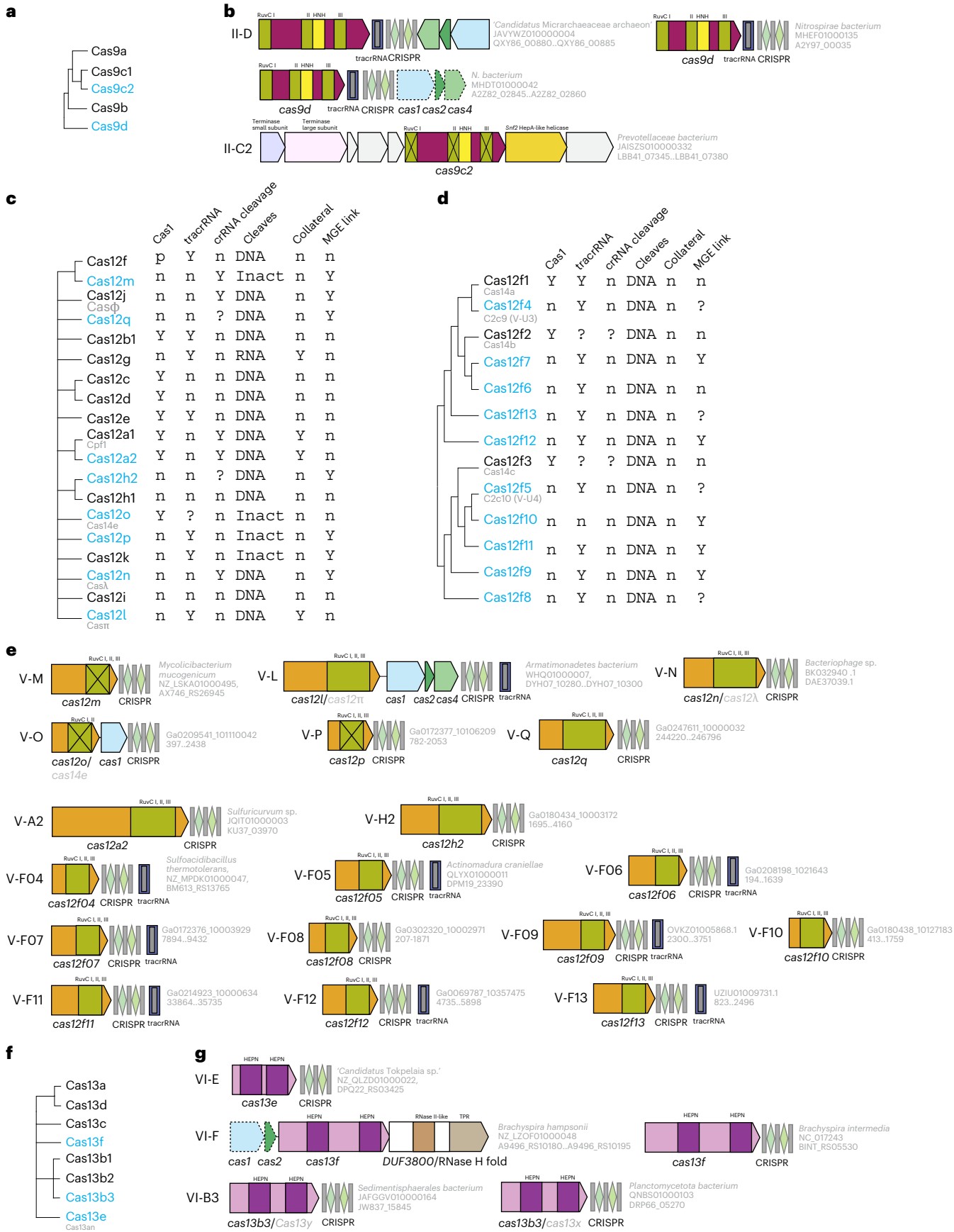

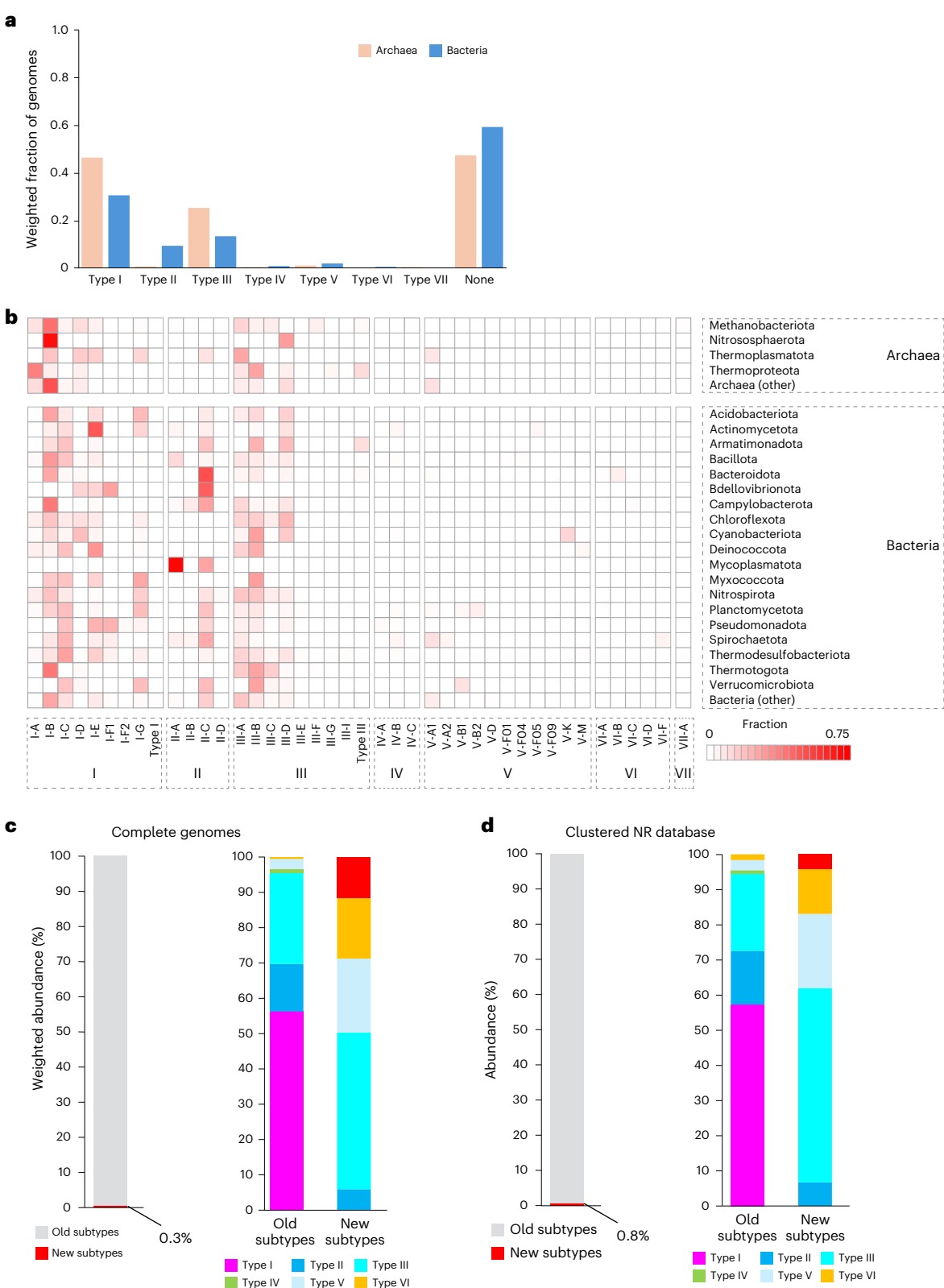

**Fig. 5 | Distribution of CRISPR–Cas systems across bacteria and archaea.**
**a**, Weighted fraction of completely sequenced archaeal and bacterial genomes encoding different types of CRISPR–Cas systems (for each taxon, the sum of genome weights across genomes, containing a system of the given type, was divided by the total sum of genome weights within the taxon). **b**, Heat map of the relative abundances of different subtypes of CRISPR–Cas systems in major clades of archaea and bacteria across completely sequenced genomes. The heat map shows the weighted fraction (scaled from 0 to 1.0) of complete CRISPR–Cas systems of a particular subtype in each

of the major archaeal and bacterial phyla. Each occurrence of a CRISPR–Cas system of a given type within a taxon was assigned a weight equal to the weight of the respective genome (details in Methods and ref. 15). The sum of the weights of the CRISPR–*cas* loci of each type was normalized to the sum of the weights across the taxon. **c**, Relative abundance of 'old' (identified in the 2020 classification[8] and data file 7 in ref. 15) and 'new' (this Analysis) subtypes of CRISPR–Cas systems in completely sequenced prokaryotic genomes and metagenomic datasets. **d**, Relative abundances of old and new (current work) subtypes of CRISPR–Cas systems in the clustered NR database.

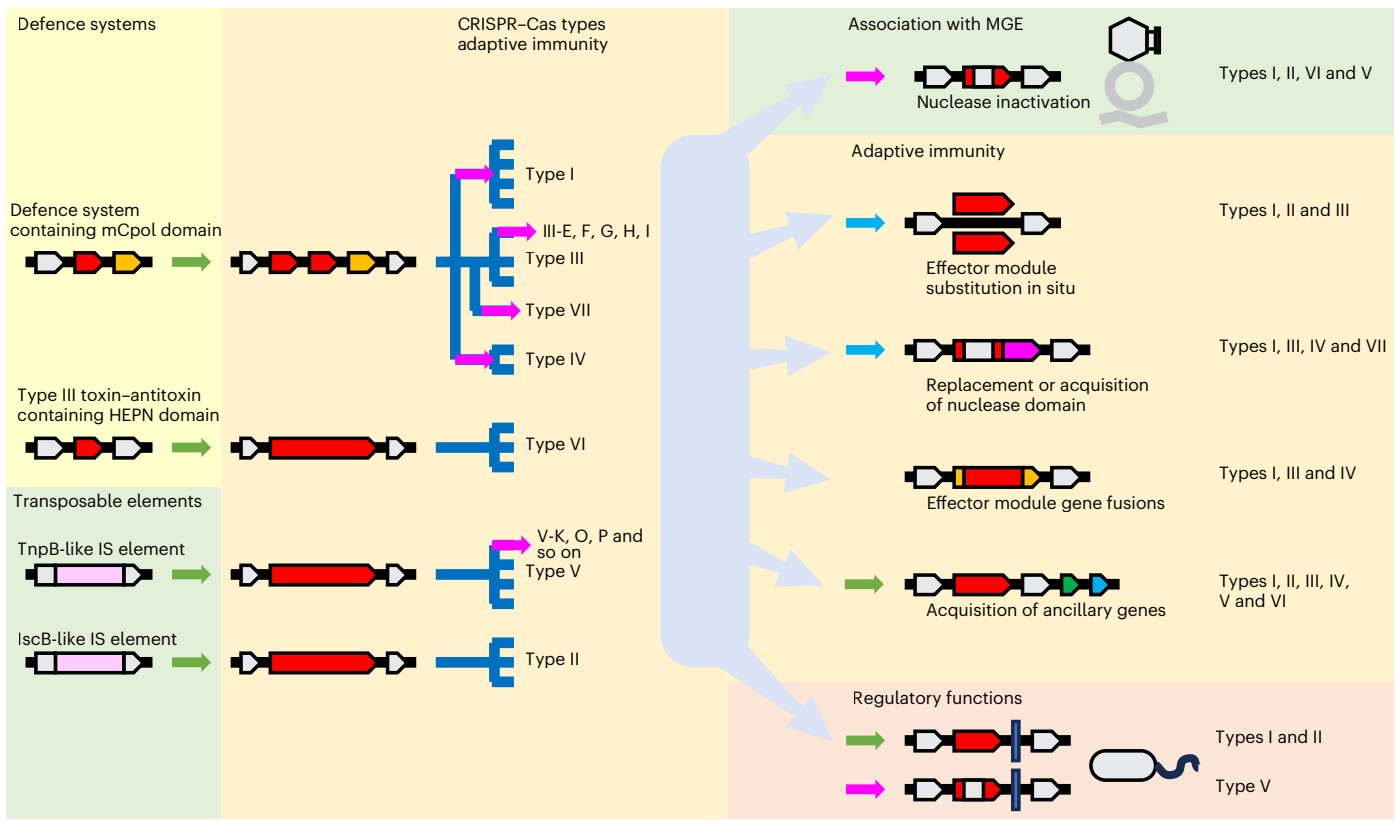

**Fig. 6 | Main trajectories and processes in the evolution of CRISPR–Cas systems.** Schematic depiction of the inferred stages of the evolution of the CRISPR–Cas systems and the evolutionary processes involved. Likely ancestral systems (left), diversification of the adaptive immunity systems (middle) and evolution of additional functions of CRISPR–Cas systems including their recruitment by MGEs (right) are illustrated. The arrows of different colours show distinct routes of evolution. IS, insertion sequences.

the 47,545 analysed genomes, complete CRISPR–Cas systems were identified in a majority of archaea (410 of 683 genomes, prevalence of 52% after adjustment for sampling bias; Methods) but in less than half of bacteria (16,488 of 46,862 genomes, 41%; Fig. 5a and data file 7 in ref. 15). As observed earlier[8], an overwhelming majority of both bacterial and archaeal thermophiles tend to possess complete CRISPR–Cas systems (weighted prevalence of 84 and 90% for thermophiles and hyperthermophiles, respectively), whereas they are substantially less prevalent in mesophiles and psychrophiles (46 and 38%, respectively). In this census, the fraction of archaeal genomes containing CRISPR–*cas* loci was much lower than that in the previous analyses. Clearly, this drop in CRISPR–Cas prevalence is due to the recently increased genome sequencing and broader sampling of mesophilic archaea, which shows that high prevalence of CRISPR–Cas is a signature of thermophiles rather than archaea as such. Among archaeal phyla, Bathyarchaeota (marine sediment mesophiles) stand out for the hitherto complete lack of detectable CRISPR–*cas* loci despite the availability of 23 diverse, completely sequenced genomes. Archaea (specifically, *Ferroglobus placidus*) harbour the only type VII system among the completely sequenced genomes. In agreement with earlier observations, the archaeal CRISPR–Cas repertoire is dominated by class 1 systems (Fig. 5b), although a few class 2 systems are scattered across Methanobacteriota and Thermoproteota, and (as pointed out earlier) subtype II-D is widely represented in the DPANN superphylum. In most archaea, type I systems comprise the majority, except in Thermoproteota where type III accounts for 51% of the CRISPR–*cas* loci. The most abundant subtype across archaeal phyla is subtype I-B, with two exceptions: subtype I-A in Thermoproteota and subtype III-A in Thermoplasmatota. The high prevalence

of subtypes III-B and III-D in Thermoproteota and Nitrososphaerota, respectively, is also notable.

Among the well-sampled bacterial phyla, only Gemmatimonadota completely lack detectable CRISPR–*cas* loci; in Chlamydiota, only one genome ('*Candidatus* Protochlamydia naegleriophila') of the 437 analysed harbours a single type I-E locus (Fig. 5b). Class 1 is also substantially more abundant than class 2 in bacteria (Fig. 5b), with a few exceptions. These include Mycoplasmatota, Bacteroidota and Bdellovibrionota, where type II systems dominate the landscape. Type V and VI systems are much less abundant across the diversity of prokaryotes than types I–IV, although type V systems are comparatively prominent in Spirochaetota, Cyanobacteriota, Planctomycetota and Verrucomicrobiota (8–13% of all CRISPR–*cas* loci), whereas type VI systems are common in Fusobacteriota and Bacteroidota (9 and 5%, respectively).

Compared with the previous census[8], the new types and subtypes identified in the intervening five years comprise only 0.3% of the CRISPR–Cas repertoire in the completely sequenced prokaryotic genomes (Fig. 5c). New subtypes of type V and type III account for most of this expansion. A qualitatively similar picture emerges from the survey of the clustered NR database (Fig. 5d). Thus, whereas all common types and subtypes of CRISPR–Cas systems seem to be already known, the discovery of new, increasingly rare, variants will probably continue in the coming years, extending the tail of the distribution.

Many CRISPR–Cas systems are currently found only in incomplete genomes or metagenomes. Not surprisingly, most of these belong to the rare and diverse type V subtypes V-C, E, G, H, I, J, L, N, O and P, and most of the new variants. Furthermore, only one representative of type VII and subtype III-I each and none of subtype III-H are present among complete genomes, whereas dozens of these loci are identifiable in

the NCBI NR database. Nevertheless, the recent massive effort on prediction of novel CRISPR–Cas systems in a large metagenomic dataset revealed only a few novel systems at the type and subtype levels[9]. It should be noted that a variety of unusual class 1 variants, such as those associated with MGE and those containing HNH nucleases fused to different Cas proteins, are difficult to identify, requiring dedicated computational approaches and manual curation to distinguish them from the mainstream CRISPR–Cas systems.

## Discussion

The updated CRISPR–Cas classification presented here consists of two classes, seven types and 46 subtypes. Because the types and subtypes of CRISPR–Cas systems that are widespread in bacteria, archaea and their MGEs seem to be already known, the current classification is likely to remain stable in its main features. By contrast, the tail of the CRISPR–Cas diversity distribution is very long and many subtypes and variants, and possibly even new types, remain to be discovered. However, these yet unknown varieties of CRISPR–Cas systems are bound to be increasingly rare and the discovery of previously unknown variants requires mining of enormous amounts of genome and metagenome sequences using dedicated computational pipelines.

The discovery and comparative analysis of previously unknown CRISPR–Cas systems have led to substantial insights into CRISPR–Cas origins and evolution. Some overarching trends are becoming apparent: (1) interplay between structural and functional complexification, and reductive, streamlining evolution of CRISPR–Cas systems; (2) tight evolutionary entanglement between CRISPR–Cas and various types of MGE[54,55], as per the 'guns for hire' concept[56]; (3) repeated exaptation of CRISPR–Cas systems for other non-defence functions[57]; (4) in situ shuffling of effector modules and (5) acquisition of ancillary genes (Fig. 6 and Extended Data Figs. 8–10; details in Supplementary Discussion). Beyond these trends, the evolutionary trajectories of class 1 and class 2 seem to be quite different and so are those of the adaptation and effector modules of the CRISPR–Cas systems.

In the near future artificial intelligence approaches will undoubtedly play an increasingly prominent role in the discovery of previously uncharacterized CRISPR–Cas variants with predefined features[58]. Such new discoveries could also come from targeted exploration of specific environments, in particular, extreme environments. Apart from new CRISPR–Cas varieties, a vast field for future studies is the widespread CRISPR–Cas exaptation, in particular, by various MGEs. Functional characterization of such repurposed CRISPR–Cas systems should help clarify fundamental aspects of co-evolution between prokaryotes and MGEs.

## Methods

### The prokaryotic genome database

A collection of 46,862 bacterial and 683 archaeal genomes available from GenBank and RefSeq that were completely sequenced or assembled at the chromosome level was downloaded in November 2023. All $75 \times 10^6$ protein sequences with unique identifiers were clustered using MMSEQS2 (ref. 59) at two levels: (1) with similarity and coverage thresholds of 0.9 and (2) with a similarity threshold of 0.5 and a coverage threshold of 0.33 (–cluster-mode 2 (greedy incremental clustering) and –cov-mode 1 (coverage calculated on the target sequence)). The optimal growth temperature was obtained for 24,711 genomes by reconciling the data downloaded from https://melnikovlab.com/gshc/ (on 3 February 2025) with genome assembly identifiers and species-level taxonomy identifiers in our dataset.

### CRISPR array detection

A total of 320,239 CRISPR arrays and CRISPR-like repetitive sequences were identified within the prokaryotic genome database using the standalone version of CRISPRCasFinder 4.2.20 (ref. 60) with default parameters, except for minimum direct repeat length of 20 base pairs.

### Protein annotation

Proteins were annotated using a PSI-BLAST[61] search against the database of all protein sequences encoded by the prokaryotic genomes in the dataset described above. Queries for these search processes included public NCBI Conserved Domain Database profiles[62] (Pfam, NCBI conserved domains (CD) and clusters of orthologous genes (COGs)), excluding public profiles for Cas proteins and including new Cas protein profiles developed in this study. A non-overlapping set of best-scoring profile hits with a PSI-BLAST $e$-value threshold of ≤0.0001 was assigned to each protein sequence.

### Phylogenetic analysis

**Genome tree.** For approximate assessment of the phylogenetic relationships between the genomes, protein sequences from the 54 nearly universal bacterial COGs and 55 nearly universal archaeal COGs were aligned using FAMSA[63]. For COGs represented by two or more paralogues, a single representative per genome with the highest similarity to the alignment consensus was selected. After removing low homogeneity and highly gapped columns[64], alignments of bacterial and archaeal proteins were concatenated separately and the corresponding approximate maximum likelihood trees were constructed using FastTree with gamma-distributed site rates and Whelan and Goldman evolutionary model[65].

**Phylogenetic trees.** For the Cas9 protein family, the following procedure was applied to facilitate the multiple sequence alignment (MSA) construction: (1) clustering using MMSEQS2 to obtain a non-redundant dataset (similarity and coverage thresholds of 0.9), (2) alignment of cluster representatives using Muscle5 (ref. 2), (3) poorly aligned sequences or fragments were discarded, (4) filtering for alignment columns with a homogeneity value of ≥0.05 and gap fraction of <0.667, and (5) the filtered alignment was used as the input for FastTree[65] to construct maximum likelihood phylogenetic tree with the Whelan and Goldman evolutionary model, gamma-distributed site rates; the same program was used to calculate support values (data file 4 in ref. 15).

To build trees for Cas1, Cas3 and the RuvC domains of Cas12/TnpB family, a modification of the above procedure was used as follows: (1) clustering using MMSEQS2 with a similarity threshold of 0.5 and a coverage threshold of 0.33, (2) alignment of sequences inside each cluster using MUSCLE5, (3) extraction of the consensus sequence from each alignment, (4) aligning the cluster consensus sequences, (5) expansion of this alignment into pseudo-MSA[66], and (6) filtering alignment and building FastTree tree as described in the previous paragraph (data file 4 in ref. 15).

**Unweighted pair group method with arithmetic mean dendrograms.** To investigate the hierarchy of similarity between protein clusters that do not readily align, relative HHSEARCH[67] scores ($S_{A,B}/\min(S_{A,A}, S_{B,B})$, where $S_{A,B}$ is the score between profiles A and B) were obtained for cluster comparison and converted to distances using the negative log transformation. Hierarchical unweighted pair group method with arithmetic mean (UPGMA) dendrograms were then constructed using scipy.cluster.hierarchy and the 'average' method in Python or hclust and the 'average' method in R. This approach was applied for Cas8/Cas10s (Extended Data Fig. 10), Cas12 (Extended Data Fig. 4b), Cas13 (Extended Data Fig. 3c), Cas7, Cas5 and Cas11 (data file 4 in ref. 15). The specific settings are indicated in the respective figure legends.

**Hybrid dendrograms.** Hybrid dendrograms were built as described previously[8]. Briefly, the FastTree program[65] (Whelan and Goldman evolutionary model, gamma-distributed site rates) was used to infer relationships within alignable clusters and the relationships between these clusters were inferred from HHalign pairwise scores using the

matrix-based UPGMA approach as described in the previous section. The FastTree trees built for clusters were grafted onto the tips of the profile similarity-based UPGMA dendrogram. Such hybrid dendrogram were built for the Cas10 and CARF families (Fig. 3a and data file 4 in ref. 15).

### Updating Cas protein profiles

Using NCBI Conserved Domain Database Cas protein profiles (Pfam, CD and COG)[62] and the previously published Cas protein profile sets[8], Cas proteins were retrieved from the prokaryotic genome database. Preliminary CRISPR–cas genomic islands were then assembled (ten genes flanking the genes cas1–cas14 from each side) for tentatively identified type I, II, III, IV and VII systems (the type V and VI systems were analysed separately). Unknown genes adjacent to known cas genes or CRISPR arrays were manually analysed using tools for detection of sequence and structural similarity, including HHPred for sequences[67] and AF2 modelling[68], followed with DALI[69] for structures. Using the retrieved Cas protein sequences and sequences from the previously published CRISPR–Cas profiles[8], sets of protein sequences were assembled for the following major protein families: Cas1, Cas2, Cas3, Cas4, Cas5, Cas6, Cas7, Cas8, Cas9, Cas10, Cas10d, Cas11, Cas14, Csx19, Csx22, Csx24, Csx25, Csx26 and CARF. Cas12 and Cas13 proteins were analysed separately as described below.

Profiles were constructed for each protein set using the following pipeline. Representatives of each protein family were selected from the 0.9 similarity clusters of the prokaryotic protein dataset (the first member of each cluster, as reported by MMSEQS2). PSI-BLAST footprints from cognate CRISPR–Cas profiles were retrieved from this set of cluster representative sequences. For manually annotated CRISPR-linked genes (unknown genes from the CRISPR–Cas islands described earlier), the entire sequence was taken as a footprint. The footprints were further clustered with a similarity threshold of 0.7 and coverage threshold of 0.9 using MMSEQ2. Profiles were generated for each cluster of the footprints according to the following procedure. The sequences within each cluster were aligned using FAMSA for clusters including >500 sequences, muscle5 with the 'super5' option for clusters including 101–500 sequences and muscle5 with the 'mpc' option for clusters including ≤100 sequences[70]. To filter fragments or partial sequences, the resulting alignment was used as a PSI-BLAST query against all cluster sequences and sequences matching <75% of the profile length were removed. The cleaned sequence set was realigned as described earlier.

Because the legacy Cas protein profiles (Pfam, CD, COG and previously constructed custom profiles[8,62]) often include inconsistently defined boundaries for the same domain, their footprints could differ strongly with respect to mapping on the protein structure. To mitigate this and obtain a consistent set of homologous sequences, the mapping and realigning procedure described above was reapplied using the cluster profile as the query for PSI-BLAST.

Each resulting 0.7 similarity cluster was used as a PSI-BLAST query against the prokaryotic database to identify hits (e-value cutoff of 0.0001) outside CRISPR–cas islands, which were treated as false positives for further calculations. A reduced set of clusters representing closely related protein groups was constructed using the 0.7 similarity cluster profiles and the hierarchical clustering procedure described earlier with a tree depth cutoff of 0.8. Profiles were generated for the resulting clusters and these clusters were used to generate a distance matrix as described earlier. Further clustering was performed using the neighbour-joining method[71] until convergence. In each neighbour-joining iteration, sequences from the closest cluster profile pair were merged, a new profile was generated with the approach described above and the distance matrix was recalculated. Clusters were merged only if the PSI-BLAST results for the merged profile contained more than 98% of the original sequences and no more than ten new false positives (except Cas3 and CARF genes, where

false-positive counts were not used for the neighbour-joining profile generation procedure given the abundance of homologues outside CRISPR–Cas systems).

The resulting clusters were manually reviewed. Clusters containing ≤3 sequences were removed if those sequences were represented in the PSI-BLAST results (e-value cutoff of 0.0001) of larger clusters. In other cases, and for CRISPR–Cas genes that are poorly represented (<3 copies) in the prokaryotic database, profiles were constructed using sequences from the clustered NR. Sequences of the respective proteins were used as PSI-BLAST queries for three iterations against the clustered NR database https://ncbiinsights.ncbi.nlm.nih.gov/2022/05/02/clusterednr_1/ to retrieve additional sequences and align them as described earlier.

Cas12 profiles were derived from distinct branches of phylogenetic trees and UPGMA dendrograms reported previously[38,39]. The Cas12b3 alignment was built using the sequences from NR. For all Cas12 families, both near-full-size alignments[38] and alignments of the RuvC-like nuclease domains alone were used to construct profiles.

Profiles for Cas13a, -c and -d were derived from previously published alignments[8]; Cas13b alignments and Cas13f were constructed in this Analysis from recently reported sequences[47,48,72,73]. For all Cas13 families, separate profiles were constructed for the N-terminal and C-terminal HEPN domains.

### Estimation of recombination within CRISPR–cas loci

A rough estimate of the extent of recombination between modules within CRISPR–cas loci was obtained by tallying the co-occurrence of cas1 with distinct effector genes (cas8, cas9, cas10 or cas12) within the same locus. If Cas1 proteins from the same strict cluster (90% identity) co-occurred with effectors from permissive clusters (50% identity), a recombination event between the adaptation module and the effector module was registered, producing a conservative estimate of inter-module swaps.

### Estimation of abundance of CRISPR–Cas systems

Profiles of large subunits of effector complexes of Class 1 systems and effector proteins of Class 2 systems were searched against the NCBI clustered NR database using MMSEQS2 (ref. 59). For most of the CRISPR effectors, the corresponding profiles were used as queries in a maximum sensitivity (−s 7.5) MMSEQS2 search with an e-value threshold of 0.0001. For type V effectors, sequences in Cas12 profiles were clustered using MMSEQS2 with similarity thresholds of 0.9, and the cluster representatives were used as queries in a maximum sensitivity (−s 7.5) MMSEQS2 search with an e-value threshold of 0.0001 and similarity threshold of 0.6 (as reported by MMSEQS2). The best non-overlapping hits that covered at least 0.75 of the query length (either profile consensus or the individual query protein) were recorded for each target protein. The number of hits was then assigned to each type and subtype, corresponding to the query provenance, and used to approximate the relative abundance of CRISPR–Cas types and subtypes in the microbial world beyond the completely sequenced genomes.

### Assembly of CRISPR–cas genomic loci

For the genes encoding all Cas proteins and cyclic oligonucleotide-based anti-phage signalling systems, genomic neighbourhoods of up ten flanking genes from each side, including predicted CRISPR arrays, were defined. Neighbourhoods containing at least one core Cas proteins (cas1–cas14) were retrieved. Cas12 proteins were identified and classified separately using Cas12/TnpB phylogenetic tree (data file 4 in ref. 15). Such analysis is required to assign subtypes and to distinguish CRISPR effectors from similar TnpB-like RNA-guided nucleases and candidate systems (collectively referred to as subtype V-U)[8,38,39]. Cas13 proteins were identified using the respective profiles and manually curated to remove false positives. Annotation of protein-coding genes with type- and subtype-specific profiles was used to classify CRISPR–Cas

systems (data files 5, 6 and 8 in ref. 15). Boundaries of CRISPR–*cas* loci were defined to include all CRISPR and Cas10-associated cyclic oligonucleotide-based anti-phage signalling systems genes and members of their respective directions at the locus margins.

For CRISPR–Cas systems poorly represented in the complete genome database used here (subtypes II-D, III-H, III-I, V-L, V-N, V-O, V-P, V-Q, VI-E, VI-F and VII), genome neighbourhoods were retrieved from the NCBI nucleotide sequence database. The anchors for these loci were obtained by searching the corresponding effector protein profiles against the NCBI clustered NR database (Cas10 for III-H, and Cas7-like and Cas10-like for III-I).

### Genome weights

To mitigate the massively uneven representation of prokaryotic clades in the complete genome database, a tree-based genome weighting scheme was used[74]. The analysis of the representation of CRISPR–Cas systems was based on weights obtained for all 47,545 complete genomes. The representation of CRISPR–Cas systems with respect to temperature preferences was analysed using the weights for the subset of 24,711 genomes with known optimal growth temperatures. For the estimation of the relative abundances of CRISPR–Cas systems, weights were derived from the subtree of the 16,898 genomes encoding at least one complete CRISPR–Cas system.

### Structural modelling and comparison

**Modelling of individual Cas proteins and extraction of core domains.** Structural models of representative Cas7, Cas8 and Cas10 proteins of different type I and III subtypes were predicted using AlphaFold2 (default parameters)[68]. Core domains of Cas8 and Cas10 (both AlphaFold2 predictions and from PDB) have been extracted by identifying fused Cas3 and Cas11 by structure comparison. To do so, full Cas8 and Cas10 structures have been compared against individual Cas3 and Cas11 structures available in PDB by foldseek. After mapping, fused Cas3, Cas5 and Cas11 domains were removed, if applicable, and the core Cas8 and Cas10 proteins kept. Core structures were verified manually afterwards and used for Cas8 and Cas10 comparison (data file 1 in ref. 15).

**Structure comparison of Cas8 and Cas10 proteins.** Core Cas8 and Cas10 domains from selected representatives of type I and III subtypes were extracted from either PDB structures or AlphaFold2 (ref. 68) models and compared all-versus-all using DALI[69] (data file 4 in ref. 15). Pairwise root mean square deviation values were used to construct an UPGMA dendrogram as follows: (1) the pairwise root mean square deviation values were normalized to the minimum of the self-scores and converted to a distance matrix on the natural logarithmic scale and (2) the UPGMA dendrogram was reconstructed from this distance matrix using the R package hclust with the argument 'method' set to average (=UPGMA).

**Structure comparison of Cas7 domains.** Individual Cas7 domains were extracted from III-I Cas7-11i (WP_207677910), III-D Cas7 (8bmw), III-E Cas7-11e (7zol) and AlphaFold2 predicted III-D2 Cas7-11c (GwCas7-11c) and compared all-versus-all by DALI. Pairwise root mean square deviation values were used to construct a UPGMA dendrogram using the R package hclust with the argument 'method' set to average (=UPGMA).

**Modelling of CRISPR effector complexes.** CRISPR effector complexes of subtypes III-G, III-H and III-I have been predicted with Alpha-Fold3[75] (default settings) using the following input sequences: III-G (NC_012623.1), *S. islandicus* Y.N.15.51, 7×Cas7_1, 1×Cas7_2, 1×Cas5, 1×Cas10, crRNA (48 nt) and crRNA (38 nt); III-H (RPGO01000026), '*Candidatus* A. ethanivorans' isolate Eth-Arch1, AEth_01085, 7×Cas7, 1×Cas5 and 1×Cas7, RNA as for III-G; III-I, (*D. magnum* WP_207677910

and WP_207677911.1), 1×Cas7-11i and 1×Cas10; and RNA from type III-E of *D. magnum* (PDB 7zol). Predicted complex structures are available in data file 1 in ref. 15.

### Criteria for identification and classification of CRISPR–*cas* systems

As in previous classifications of CRISPR–Cas systems, the main approach for their identification was the search of protein sequences encoded in bacterial and archaeal genomes using position-specific iterated BLAST (PSI-BLAST)[61]. The queries for these searches were position-specific scoring matrices (PSSMs) generated from multiple alignments of the 14 core Cas proteins and additional ancillary proteins strongly associated with CRISPR–Cas systems. The use of PSSMs rather than individual protein sequences as queries is essential for reliable identification of CRISPR–Cas systems due to the typically high evolution rate of *cas* genes[76]. Together, for this work, we assembled a set of 915 PSSMs covering *cas* and other associated genes including 293 newly constructed and curated ones that were used to identify 146,993 core *cas* genes (*cas1*–*cas14*) in 47,545 completely sequenced bacterial and archaeal genomes that were available at the NCBI in November 2023 (Fig. 1a, 'The prokaryotic genome database' section and data files 5, 6 and 8 in ref. 15). For most of the core Cas protein families, multiple PSSMs had to be used because high sequence divergence did not allow reliable recognition with a single PSSM. For all core genes, similarity dendrograms or phylogenetic trees were constructed to assess the relationships among the Cas proteins in the respective families (data file 4 in ref. 15). In addition, representatives of several subtypes and variants of type V systems were missing in the analysed genome collection, so the sequences were extracted from recent publications and used as queries to search the GenBank ('Protein annotation' section). After extensive manual curation of the results, this search yielded 21,474 complete (that is, those containing all core as well as subtype-specific signature genes of the effector module) CRISPR–Cas systems that were assigned to types and subtypes (Extended Data Figs. 1–4 and data file 8 in ref. 15). The loci that could not be confidently assigned to any of the previously identified subtypes were designated representatives of new subtypes or variants. New variants identified in recent genome and metagenome analyses but missing in the complete genome collection were added to the list of CRISPR–*cas* loci (Extended Data Figs. 1–4 and data file 8 in ref. 15). The substantially updated collection of multiple alignments and PSSMs for Cas protein families that was developed and employed in this work is available in data files 5 and 6 in ref. 15. These alignments and/or PSSMs can be used as tools for identification and classification of CRISPR–*cas* loci in sequenced genomes and metagenomes. Several conflicts with the published classification of CRISPR–Cas systems are listed in Supplementary Table 3.

### Reporting summary

Further information on research design is available in the Nature Portfolio Reporting Summary linked to this article.

## Data availability

All the data used and generated in this work are available at https://doi.org/10.5281/zenodo.17388109 (ref. 15).

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

## Acknowledgements

K.S.M., S. A. Shmakov, Y.I.W., P.M., D.H.H. and E.V.K. are supported through the Intramural Research Program of the National Institutes of Health of the USA (National Library of Medicine). S. A. Shmakov is supported by funding of the Basic Research Program by the National Research University Higher School of Economics. C.L.B. is supported by European Research Council grants 865973 and 101158249. D.C. and J.T. are supported by Arbor Biotechnologies. P.H. is supported by IFF. S.M. was supported by funding from the Natural Sciences and Engineering Research Council of Canada (Discovery program) and holds a Tier 1 Canada Research Chair in Bacteriophages. F.J.M.M. is supported by the grants PID2023-150750NB-I00 (funded by MICIU/AEI/10.13039/501100011033 and by ERDF/EU), PROMETEU/2021/057 (funded by Conselleria d'Educació, Cultura, Universitats i Ocupació, Generalitat Valenciana, Spain) and INNEST/2024/427 (funded by Agencia Valenciana de Innovación—IVACE+I Innovación—and by the European Union through the ERDF Program of the Valencian Community 2021–2027). P.P. receives funding from the EMBC under EMBO Installation Grant (5342-2023) and is supported by the LMT under the 'University Excellence Initiatives' (measure number 12-001-01-01-01, project S-A-UEI-23-10). R.P.-R. is supported by a Lundbeck Fonden grant (R347-2020-2346) and a research grant from VILLUM FONDEN (VIL60763). S. A. Shah is a recipient of a Novo Nordisk Foundation project grant in basic bioscience (NNF18OC0052965). C.V. is supported by intramural funds of the Vilnius University. M.P.T. is supported by the National Institutes of Health (grant R35GM118160). A.F.Y. is supported by an Environmental Biotechnology Innovation Centre (EBIC), UKRI Engineering Biology Mission Hub grant (BB/Y008332/1). F.Z. is supported by Howard Hughes Medical Institute; Yang Tan Collective; Broad Institute Programmable Therapeutics Gift Donors; Pershing Square Foundation, William Ackman, and Neri Oxman; Phillips family; J. and P. Poitras; B.T. Charitable Trust. R. Backofen is supported by the German Research Foundation (SFB 1597/1 'Small Data').

## Author contributions

K.S.M., S. A. Shmakov, Y.I.W., P.M., P.P. and E.V.K. researched the data for the article. K.S.M., Y.I.W. and E.V.K. wrote the article. K.S.M., S. A. Shmakov, Y.I.W., P.M., H.A-T., C.L.B., S.J.J.B., E.C., D.C., J.D., D.H.H., P.H., S.M., F.J.M.M., P.P., R.P.-R., S. A. Shah, V.S., M.P.T., J.T., Č.V., M.F.W., A.F.Y., F.Z., R.A.G., R. Backofen, J.v.d.O., R. Barrangou and E.V.K substantially contributed to the discussion of the content, edited and approved the paper.

## Competing interests

The authors declare no competing interests.

## Additional information

**Extended data** is available for this paper at https://doi.org/10.1038/s41564-025-02180-8.

**Correspondence and requests for materials** should be addressed to Eugene V. Koonin.

 

[1]Division of Intramural Research, National Library of Medicine, National Institutes of Health, Bethesda, MD, USA. [2]National Research University Higher School of Economics, Moscow, Russia. [3]Institute for Protein Design, University of Washington, Seattle, WA, USA. [4]Helmholtz Institute for RNA-based Infection Research (HIRI), Helmholtz Centre for Infection Research (HZI), Würzburg, Germany. [5]Medical Faculty, University of Würzburg, Würzburg, Germany. [6]Kavli Institute of Nanoscience, Department of Bionanoscience, Delft University of Technology, Delft, The Netherlands. [7]Max Planck Unit for the Science of Pathogens, Humboldt University, Berlin, Germany. [8]Arbor Biotechnologies, Cambridge, MA, USA. [9]Department of Chemistry, University of California, Berkeley, CA, USA. [10]Innovative Genomics Institute, University of California, Berkeley, CA, USA. [11]Department of Molecular and Cell Biology, University of California, Berkeley, CA, USA. [12]California Institute for Quantitative Biosciences, University of California, Berkeley, CA, USA. [13]Gladstone Institute of Data Science and Biotechnology, Gladstone Institutes, San Francisco, CA, USA. [14]Howard Hughes Medical Institute, University of California, Berkeley, CA, USA. [15]National Center for Biotechnology Information, National Library of Medicine, National Institutes of Health, Bethesda, MD, USA. [16]IFF, Dangé-Saint-Romain, France. [17]Département de biochimie, de microbiologie et de bio-informatique, Faculté des sciences et de génie, Université Laval, Québec City, Quebec, Canada. [18]Departamento de Fisiología, Genética y Microbiología, Universidad de Alicante, Alicante, Spain. [19]LSC-EMBL Partnership Institute for Genome Editing Technologies, Life Sciences Center, Vilnius University, Vilnius, Lithuania. [20]Section of Microbiology, Department of Biology, University of Copenhagen, Copenhagen, Denmark. [21]Copenhagen Prospective Studies on Asthma in Childhood (COPSAC), Herlev and Gentofte Hospital, University of Copenhagen, Gentofte, Denmark. [22]Institute of Biotechnology, Life Sciences Center, Vilnius University, Vilnius, Lithuania. [23]Biochemistry and Molecular Biology, Genetics and Microbiology, University of Georgia, Athens, GA, USA. [24]Biomedical Sciences Research Complex, University of St. Andrews, St. Andrews, UK. [25]Department of Chemical Engineering and Applied Chemistry, University of Toronto, Toronto, Ontario, Canada. [26]Centre for Environmental Biotechnology, School of Natural Sciences, Bangor University, Bangor, UK. [27]Broad Institute of MIT and Harvard, Cambridge, MA, USA. [28]McGovern Institute for Brain Research, Massachusetts Institute of Technology, Cambridge, MA, USA. [29]Howard Hughes Medical Institute, Cambridge, MA, USA. [30]Department of Brain and Cognitive Sciences, Massachusetts Institute of Technology, Cambridge, MA, USA. [31]Yang Tan Collective, Massachusetts Institute of Technology, Cambridge, MA, USA. [32]Department of Biological Engineering, Massachusetts Institute of Technology, Cambridge, MA, USA. [33]Department of Stem Cell and Regenerative Biology, Harvard University, Cambridge, MA, USA. [34]Archaea Centre, Department of Biology, Copenhagen University, Copenhagen, Denmark. [35]BIOSS Centre for Biological Signaling Studies, Cluster of Excellence, University of Freiburg, Freiburg, Germany. [36]Laboratory of Microbiology, Wageningen University, Wageningen, The Netherlands. [37]Department of Food, Bioprocessing, and Nutrition Sciences, North Carolina State University, Raleigh, NC, USA. [38]These authors contributed equally: Kira S. Makarova, Sergey A. Shmakov. ✉e-mail: koonin@ncbi.nlm.nih.gov

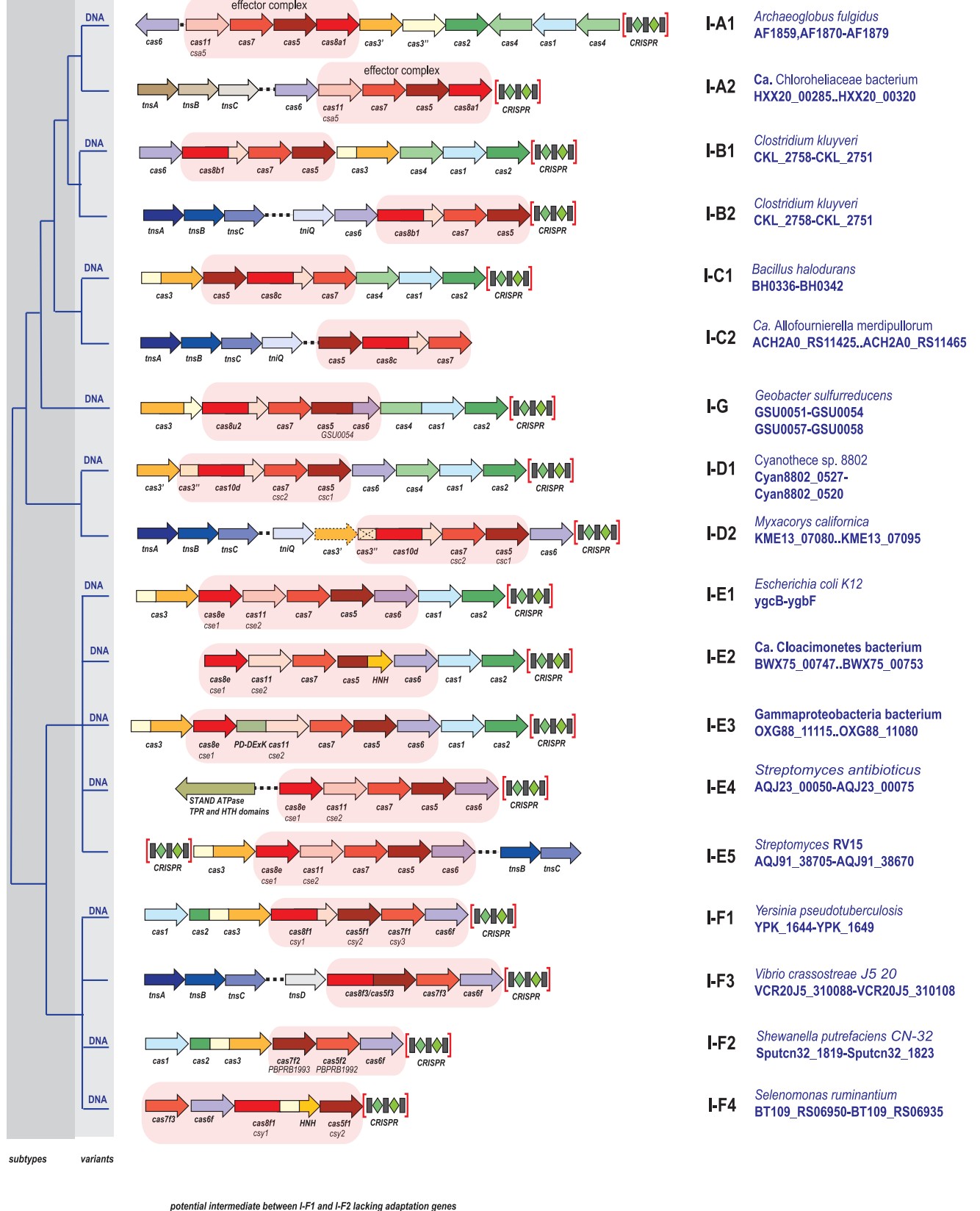

**Extended Data Fig. 1 | See next page for caption.**

**Extended Data Fig. 1 | Updated classification of type I CRISPR–Cas systems.**
The figure schematically shows representative (typical) CRISPR–cas loci for each class 1 subtype and selected distinct variants, with the dendrogram on the left showing the likely evolutionary relationships between the types and subtypes. The column on the right indicates the organism and the corresponding gene range. Homologous genes are colour-coded and identified by a family name.

The gene names follow the previous classification[8]. The pink shading shows the effector module. The grey shading of different hues shows the two levels of classification, subtypes and variants. Where both a systematic name and a legacy name are commonly used, the legacy name is given under the systematic name. DNA and RNA are the molecules targeted by the CRISPR–Cas systems.

## Type III

**III-A1** *Staphylococcus epidermidis* SERP2463-SERP2455

**III-A2** *Clostridium_scindens* HDCHBGLK_RS07485-HDCHBGLK_RS07505

**III-D1** *Synechocystis sp.* 6803 sll7067-sll7063

**III-D2** *groundwater metagenome* OBEQ011807420.1, PF1131-PF1124

**III-E** *Candidatus Scalindua brodae* SCABRO_02601,SCABRO_02597 SCABRO_02593,SCABRO_02595

**III-I** *Desulfonema magnum* dnm_RS15925..dnm_RS15930

**III-H** *Ca.* Argoarchaeum ethanivorans AEth_01085..AEth_01088

**III-F** *Thermotoga lettingae* TMO Tlet_0097-Tlet_0100

**III-C** *Methanothermobacter thermautotrophicus* MTH328-MTH323

**III-B** *Pyrococcus furiosus* PF1131-PF1124

**III-G** *Sulfolobus islandicus* Y.N.15.51 2168177..2173400

## Type IV

**IV-A1** *Thioalkalivibrio sp.* K90mix TK90_2699-TK90_2703

**IV-A2** *Alphaproteobacteria bacterium* US3C007 UM181_08495-UM181_08515

**IV-B** *Rhodococcus jostii* RHA1 RHA1_ro10069-RHA1_ro10072

**IV-C** *Thermoflexia bacterium* D6793_05715-D6793_05700

*Cohnella kolymensis* VKM SD71_04535-SD71_04555

*Halomonas sp. MES3-P3E* CXF87_RS00955..CXF87_RS00990

Hybrid system

Potential IV-A CAST

## Type VII

**VII-A** *Ferroglobus placidus DSM 10642* Ferp_0587..Ferp_0589

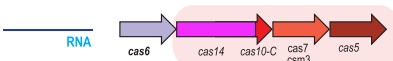

**Extended Data Fig. 2 | See next page for caption.**

**Extended Data Fig. 2 | Updated classification of CRISPR types III, IV and VII CRISPR.** Designations are the same as in Extended Data Fig. 1. Additional subunits of effector complexes are shown as grey arrows. Most of the subtype III-B, III-C, III-E, III-F loci as well as IV-B and IV-C loci lack CRISPR arrays and are shown accordingly although for each of the type III subtypes exceptions have been detected. Dashed line leading to type III-E indicates its likely origin from III-D2. Abbreviations: CHAT, protease domain of the caspase family; TPR, Tetratricopeptide repeats; RT, reverse transcriptase. DNA and RNA are the molecules targeted by the systems.

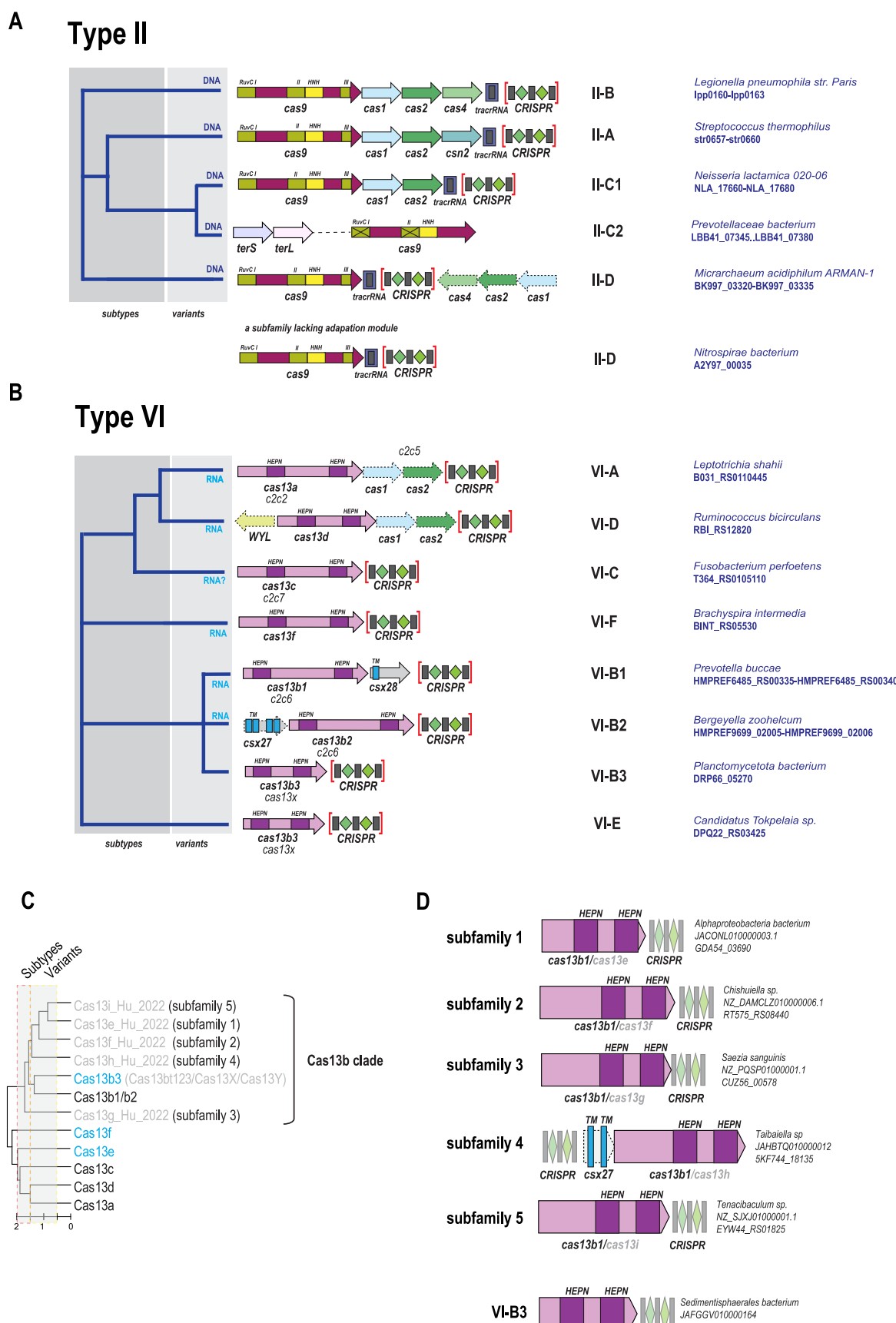

**Extended Data Fig. 3 | See next page for caption.**

**Extended Data Fig. 3 | Updated classification of CRISPR types II and VI.**
**a,b**, The figure schematically shows representative (typical) CRISPR–cas loci of each type II (**a**) and type VI subtype (**b**) and selected distinct variants, with the dendrogram on the left showing the likely evolutionary relationships between the types and subtypes. The column on the right indicates the organism and the corresponding gene range. Homologous genes are colour-coded and identified by a family name following the previous classification[8]. Where both a systematic name and a legacy name are commonly used, the legacy name is given under the systematic name. The grey shading of different hues shows the two levels of classification, subtypes and variants. The adaptation module genes cas1 and cas2 are present in only a subset of the subtype VI-A and VI-D loci and are accordingly shown by dashed lines. The WYL-domain-encoding genes and csx27 genes are also dispensable and thus shown by dashed lines. Additional genes encoding components of the interference module, such as tracrRNA, are shown. The domains of the effector proteins are colour-coded: RuvC-like nuclease, green;

HNH nuclease, yellow; HEPN RNase, purple; transmembrane domains, blue. DNA and RNA are the molecules targeted by the systems. **c**, Deep relationships among type VI effector families. Profile–profile comparisons were performed and the UPGMA dendrogram was constructed as described in Methods. The tree is based on the most conserved C-terminal HEPN domain alignments only. HHsearch was run with the minimum length coverage for hits set to l = 0.33, -u = 2.3 -gcut = 0.667. Multiple alignments (profiles) of the C-terminal HEPN domains are available in data file 5 in ref. 15, and the original tree is available in data file 4 in ref. 15. The dashed line corresponds to the tree depth D between 1.5 and 2 (D = 2 roughly corresponds to the pairwise HHsearch similarity score of $\exp(2D) \approx 0.02$ relative to the self-score) and separates most of the subtypes assigned previously or in this work. New subtypes are highlighted by blue. **d**, Organization of representative loci for distinct variants of subtype VI-B. Designations are the same as in **b**.

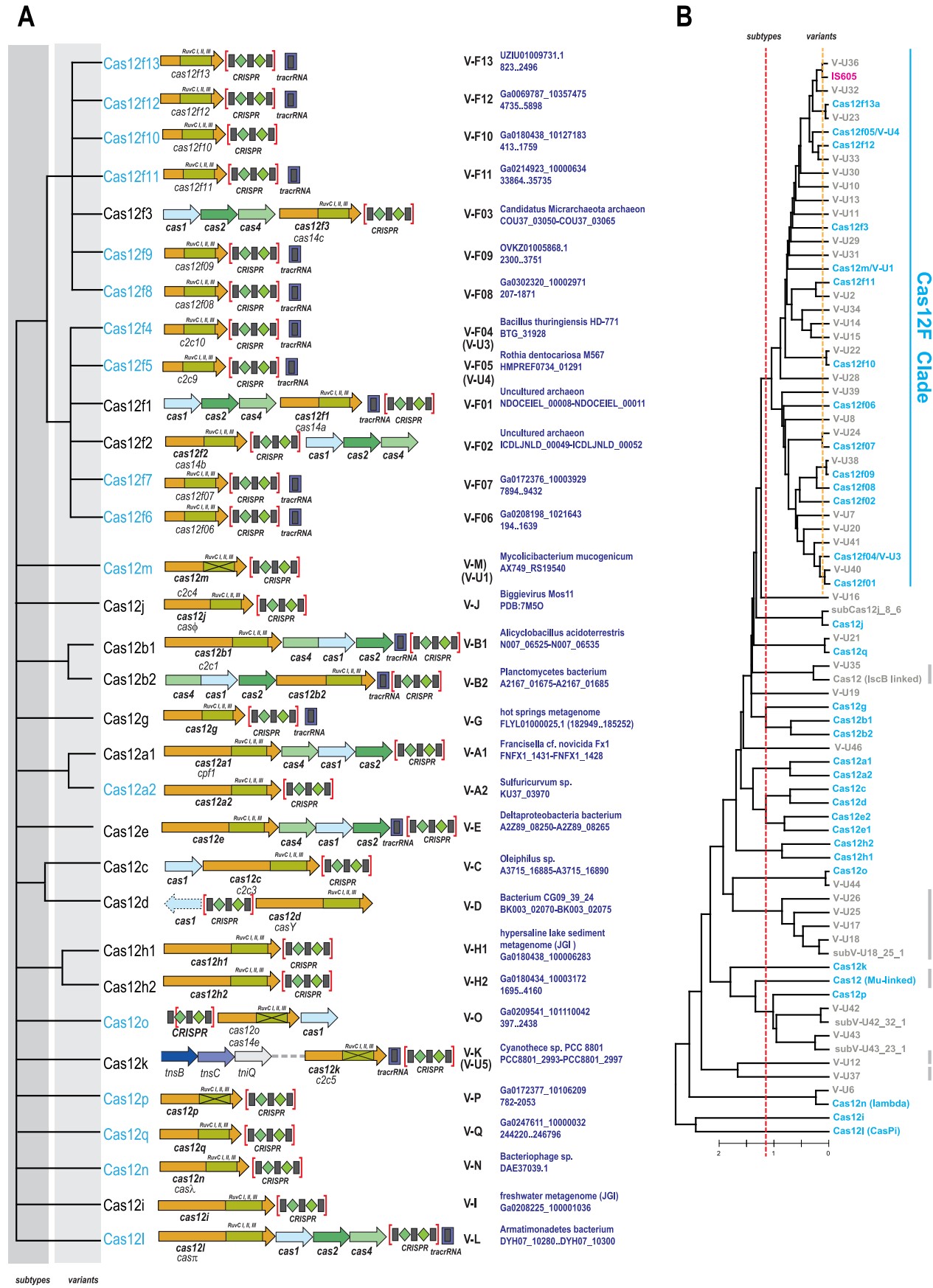

**Extended Data Fig. 4 | See next page for caption.**

**Extended Data Fig. 4 | Updated classification of type V CRISPR–Cas systems. a**, Schematics of organization of type II CRISPR–Cas systems. The figure schematically shows representative (typical) CRISPR–cas loci of each experimentally characterized and/or described in previous classification type V subtypes and distinct variants, with the dendrogram on the left showing the likely evolutionary relationships between the types and subtypes. The column on the right indicates the organism and the corresponding gene range. Homologous genes are colour-coded and identified by a family name following the previous classification[8]. Where both a systematic name and a legacy name are commonly used, the legacy name is given under the systematic name. The grey shading of different hues shows the two levels of classification, subtypes and variants. The adaptation module genes *cas1* and *cas2* are present in only a subset of the type V subtypes. Dispensable (and/or missing in some subtypes and variants) components are indicated by dashed outlines. Additional genes encoding components of the interference module, such as tracrRNA, are shown. The domains of the effector proteins are colour-coded: RuvC-like nuclease (RuvC motifs I, II, III), green. **b**, Deep relationships between type V effector families. Profile–profile comparisons (RuvC-domain only) were performed and the UPGMA dendrogram was constructed as described in Methods. Specifically, HHsearch was run with the minimum length coverage for hits set to 0.033, -u = 2.3 -gcut = 0.667. Multiple alignments (profiles) used for this analysis are available in data file 5 in Ref. [15] and the original tree is available in data file 4 in ref. [15]. The dashed line corresponds to the tree depth D between 1.5 and 2 (D = 2 roughly corresponds to the pairwise HHsearch similarity score of $\exp(2D) \approx 0.02$ relative to the self-score) and separates most of the subtypes that were previously assigned previously or in this work. New subtypes are highlighted by blue colour. IS605 (magenta) stands for TnpB RNA-guided nuclease encoded by IS605 family transposons and not associated with CRISPR arrays.

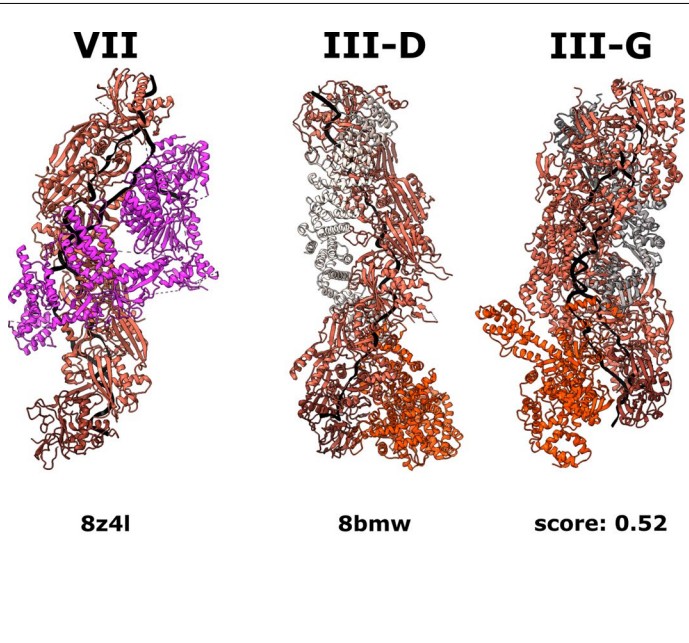

**VII**  
8z4l

**III-D**  
8bmw

**III-G**  
score: 0.52

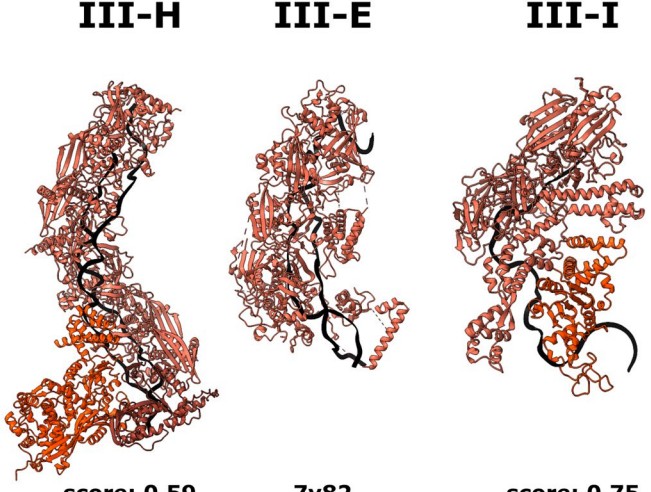

**III-H**  
score: 0.59

**III-E**  
7y82

**III-I**  
score: 0.75

**Extended Data Fig. 5 | Alphafold 3 models for III-G, III-H, III-I CRISPR effector complexes compared with solved structures of III-D1, III-E and VII effector complexes**. The models are the same as schematically shown in Fig. 2b. Distinct subunits are coloured as in Fig. 2b.

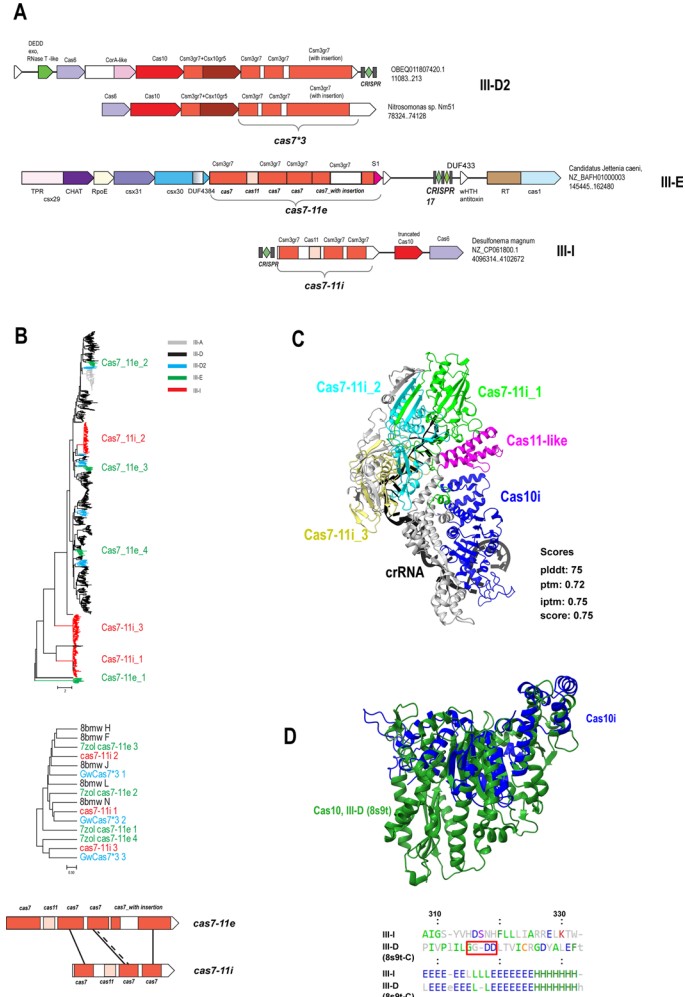

**Extended Data Fig. 6 | CRISPR–Cas subtype III-I. a**, Comparison of representative typical III-D2, III-E and III-I loci organization. Cas7 and Cas11 domains within large multidomain proteins are shown by boxes within the respective arrows. **b**, Dendrograms for individual Cas7 domains. The larger tree is a hybrid UPGMA/FastTree tree built for all best hits obtained by PSI-BLAST search with the three individual Cas7 domain of Cas7-11i used as queries. The smaller UPGMA dendrogram was built using a matrix of Pairwise rmsd scores as obtained by DALI comparison for individual Cas7 domain from the Cas7-11i AF3 model, Cas7-11e structure (7zol), AF3 model for GwCas7*3 28 and the Cas7 domain from the III-D effector complex structure (8bmw). Underneath the trees, a scheme of similarity between Cas7 domain of Cas7-11i and Cas7-11e are shown. Solid line indicates Cas7 domains with structural similarity and dashed line shows domains with significant sequences similarity. **c**, AF3 model for Cas7-11i (WP_207677910.1) and Cas10i (WP_207677911.1) complexed with crRNA from subtype III-D. Cas7 and putative Cas11 domain are coloured, Cas10i is shown in blue. **d**, Structural alignment of AF2 model of Cas10i and Cas10d structure (8s9t-C). Below the DALI structure-guided alignment, the alignment of the catalytic motif (GGDD) of Cas10 and the corresponding region of Cas10i, demonstrating the disruption of the catalytic site in the latter.

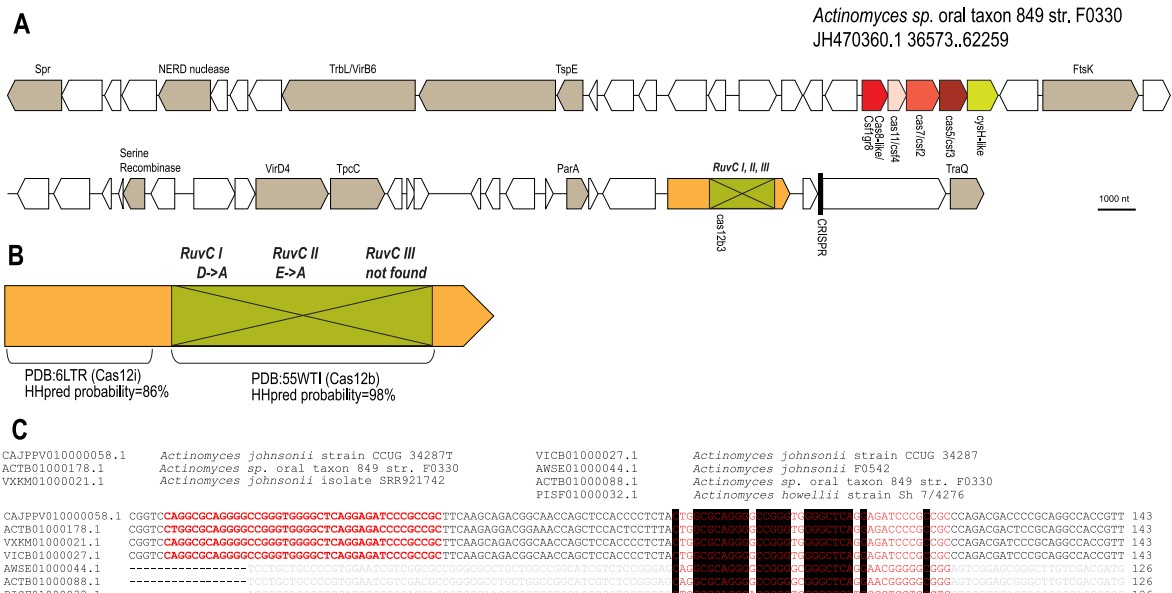

**Extended Data Fig. 7 | Inactivated type V-B variant Cas12b3. a**, Genetic organization of Actinomyces sp. conjugative plasmid region encoding type IV-B and the B3 variant of subtype V-B. Plasmid-related genes are shown in brown. Other genes (black) are mostly uncharacterized or unrelated to CRISPR or known plasmid genes. **b**, Schematic representation of HHpred search results and substitution of key amino acids of the RuvC-I and RuvC-II sites in Cas12b3. **c**, Multiple alignment of mini CRISPR array associated with the V-B3 variant. CRISPR repeats are shown in red. Genome names and accession numbers are indicated above the alignment.

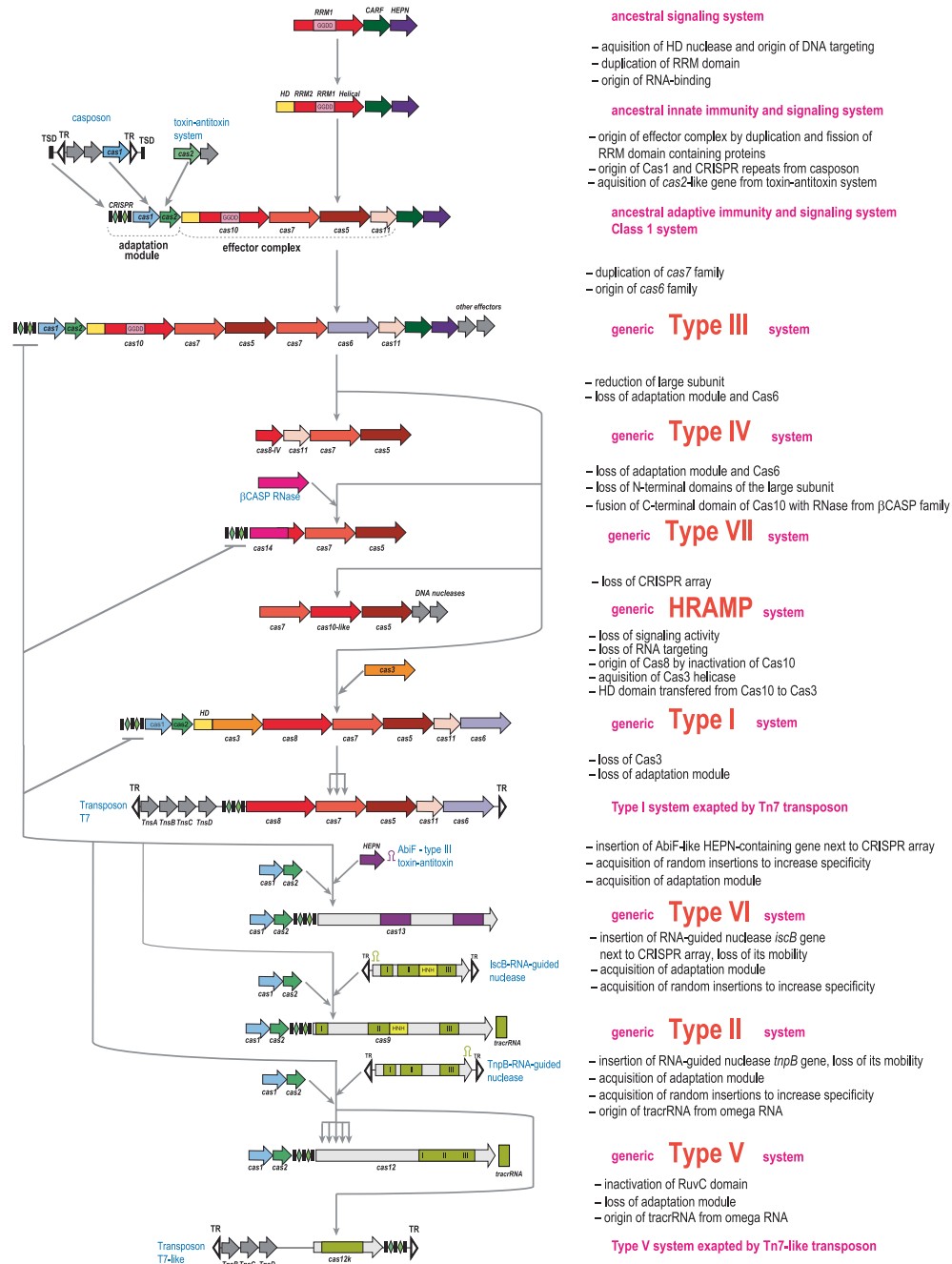

**Extended Data Fig. 8 | Hypothetical scenario for the origins and evolution of CRISPR–Cas systems.** The figure is an amended version of Fig. 6 from the 2020 classification of CRISPR–Cas systems[8]. The key evolutionary events are described to the right of the images. The multiforking arrows denote events that have been inferred to have occurred on multiple, independent occasions during the evolution of CRISPR–Cas systems. Additional abbreviations: "GGDD", key catalytic motif of the cyclase/polymerase domain of Cas10 that is involved in the synthesis of cOA; TR, terminal repeats.

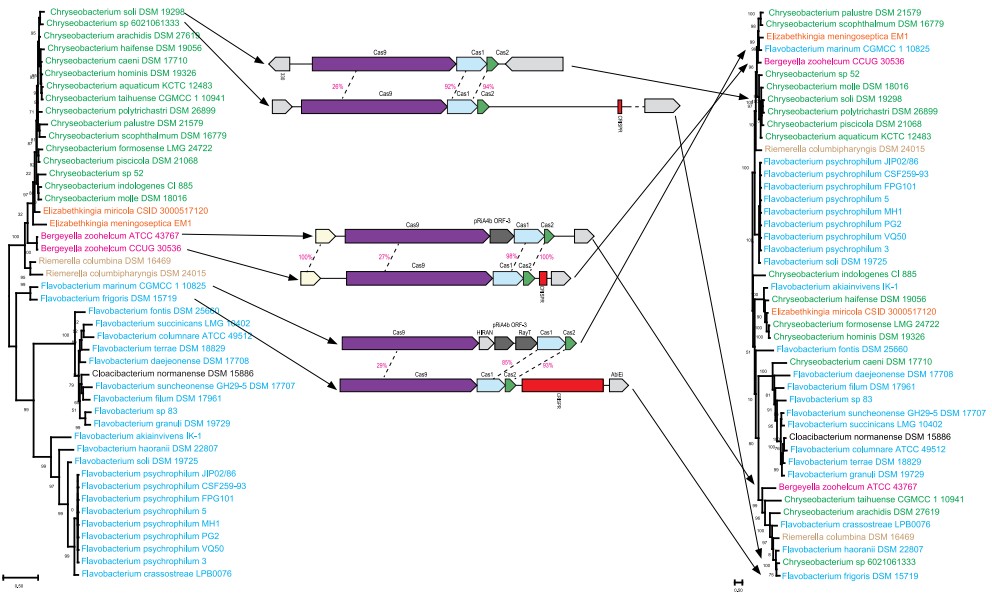

**A**

**B**

| System | No. of distinct adaptation modules involved in recombination | No. of distinct adaptation modules in *cas* loci | Fraction of distinct adaptation modules involved in recombination |
|---|---|---|---|
| Type I | 175 | 12008 | 1.5% |
| Type II | 36 | 3324 | 1.1% |
| Type III | 23 | 1315 | 1.7% |
| Type V | 0 | 87 | 0% |
| Type VI | 0 | 4 | 0% |

**Extended Data Fig. 9 | Example of Cas9 shuffling in situ and estimate of the shuffling frequency. a**, Phylogenetic analysis of subtype II-C genes from Flavobacteriales. Cas1 phylogenetic tree is shown on the left and Cas9 tree is shown on the right. Both trees were constructed using FastTree as described in Methods. Species of different genera are shown in different colours. Arrows indicate several outstanding examples of cas9 exchanged in situ (when closely related cas1 genes are associated with distantly related cas9 genes). The loci schematics and percent identity for selected genes are shown. **b**, Estimated shuffling rate of adaptation versus effector genes.

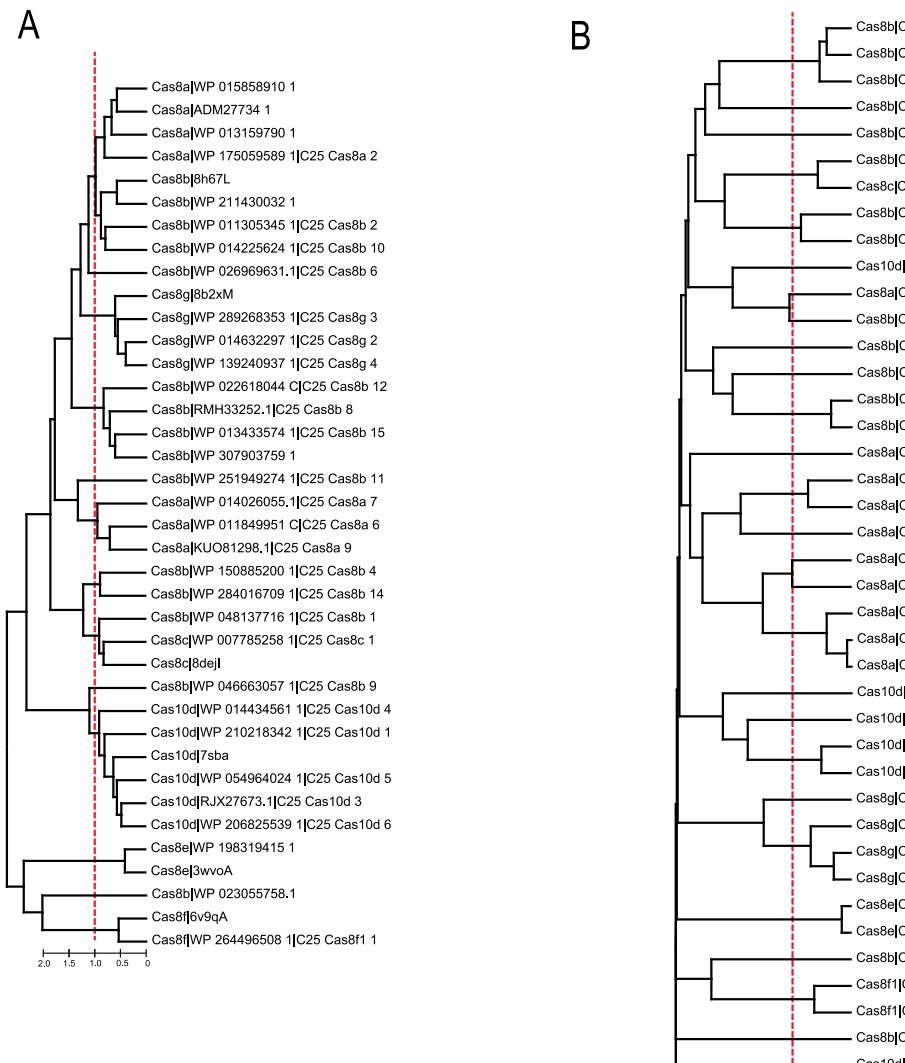

**Extended Data Fig. 10 | Deep relationships among structures of large subunits of type I effector c complexes. a**, Relationships between the structures of type I large subunit representatives (Cas8 and Cas10d families). Cas8 and Cas10d correspond to distinct profiles/subfamilies (sdata file 5 and 6 in ref. 15), and representatives of each subfamily was modelled with AF2 (ref. 68; with max_template_date = 2024-12-12). In addition, resolved structures of Cas8 and Cas10d were retrieved from PDB[77]. Additional domains present in some of the Cas8 and Cas10d proteins (such as Cas5, Cas3 and Cas11) were identified and trimmed off the structure to keep the respective core structures. These core structures were compared all against all using DALI[69]. Pairs without detectable similarity (no Dali z-score reported) were set artificially to a z-score of 0.1. The pairwise DALI z-scores were normalized by the minimum of the self-scores and converted to a distance matrix on the natural log scale. The UPGMA dendrogram was reconstructed from this distance matrix. A depth of -1 (red dashed line) corresponds to a pairwise z-score of -7.5−9. Profile IDs are indicated after the vertical bar. **b**, Profile–profile comparisons for large subunits of type I effector complexes (Cas8 and Cas10d families) were performed and the UPGMA dendrogram was constructed as described in Methods. HHsearch was run with the minimum length coverage for hits set to l = 0.33, -u = 2.3 -gcut = 0.667. Multiple alignments (profiles) used for this analysis are available in data files 5 and 6 in ref. 15. The dashed line corresponds to the tree depth D = 2, roughly corresponding to the pairwise HHsearch similarity score of exp(2D) ≈ 0.02 relative to the self-score. Typically, this tree depth reflects reliable sequence similarity.

# Reporting Summary

## Statistics

For all statistical analyses, confirm that the following items are present in the figure legend, table legend, main text, or Methods section.

| n/a | Confirmed | |
|---|---|---|
| ☐ | ☒ | The exact sample size (*n*) for each experimental group/condition, given as a discrete number and unit of measurement |
| ☒ | ☐ | A statement on whether measurements were taken from distinct samples or whether the same sample was measured repeatedly |
| ☒ | ☐ | The statistical test(s) used AND whether they are one- or two-sided<br>*Only common tests should be described solely by name; describe more complex techniques in the Methods section.* |
| ☒ | ☐ | A description of all covariates tested |
| ☒ | ☐ | A description of any assumptions or corrections, such as tests of normality and adjustment for multiple comparisons |
| ☒ | ☐ | A full description of the statistical parameters including central tendency (e.g. means) or other basic estimates (e.g. regression coefficient) AND variation (e.g. standard deviation) or associated estimates of uncertainty (e.g. confidence intervals) |
| ☒ | ☐ | For null hypothesis testing, the test statistic (e.g. *F*, *t*, *r*) with confidence intervals, effect sizes, degrees of freedom and *P* value noted<br>*Give P values as exact values whenever suitable.* |
| ☒ | ☐ | For Bayesian analysis, information on the choice of priors and Markov chain Monte Carlo settings |
| ☒ | ☐ | For hierarchical and complex designs, identification of the appropriate level for tests and full reporting of outcomes |
| ☒ | ☐ | Estimates of effect sizes (e.g. Cohen's *d*, Pearson's *r*), indicating how they were calculated |

*Our web collection on statistics for biologists contains articles on many of the points above.*

## Software and code

Policy information about availability of computer code

| Data collection | Publicly available sequence data from NCBI GenBank was downloaded and accessed between November 2023 and March 2025 using common FTP and Web interface. |
|---|---|
| Data analysis | MMseqs2 v. b22d5f6d02cb27ebc2cd931d8d20fe92ff54b8a8; CRISPRCasFinder v. 4.2.20; psiblast v. 2.16.1+; FAMSA v. 2.2.2-7eb7612; FastTree v. 2.1.4 SSE3; MUSCLE5 5.0.1278_linux64; HHsearch v. 3.0.3; FastME v. 2.1.6.2; AlphaFold2; AlphaFold3 |

For manuscripts utilizing custom algorithms or software that are central to the research but not yet described in published literature, software must be made available to editors and reviewers. We strongly encourage code deposition in a community repository (e.g. GitHub). See the Nature Portfolio guidelines for submitting code & software for further information.

## Data

Policy information about availability of data

All manuscripts must include a data availability statement. This statement should provide the following information, where applicable:
- Accession codes, unique identifiers, or web links for publicly available datasets
- A description of any restrictions on data availability
- For clinical datasets or third party data, please ensure that the statement adheres to our policy

Supplementary information for this paper is available at: https://ftp.ncbi.nih.gov/pub/wolf/_suppl/CRISPRclass25/ and https://doi.org/10.5281/zenodo.15085843

# Research involving human participants, their data, or biological material

Policy information about studies with human participants or human data. See also policy information about sex, gender (identity/presentation), and sexual orientation and race, ethnicity and racism.

| | |
|---|---|
| Reporting on sex and gender | n/a (no human participants) |
| Reporting on race, ethnicity, or other socially relevant groupings | n/a |
| Population characteristics | n/a |
| Recruitment | n/a |
| Ethics oversight | n/a |

Note that full information on the approval of the study protocol must also be provided in the manuscript.

# Field-specific reporting

Please select the one below that is the best fit for your research. If you are not sure, read the appropriate sections before making your selection.

☒ Life sciences ☐ Behavioural & social sciences ☐ Ecological, evolutionary & environmental sciences

For a reference copy of the document with all sections, see nature.com/documents/nr-reporting-summary-flat.pdf

# Life sciences study design

All studies must disclose on these points even when the disclosure is negative.

| | |
|---|---|
| Sample size | All completely sequenced prokaryotic genomes available in November 2023 were used in the analysis without subsampling. |
| Data exclusions | No data were excluded. |
| Replication | All results are produced using computational analyses, relying on algorithms that do not display an inherently indeterministic behavior. Thus they are reproducible up to the data and software version stability. |
| Randomization | Randomization is not relevant to this study because complete available datasets were analyzed. |
| Blinding | Blinding is not relevant to this study because these computational analyses are not subject to human biases. |

# Reporting for specific materials, systems and methods

We require information from authors about some types of materials, experimental systems and methods used in many studies. Here, indicate whether each material, system or method listed is relevant to your study. If you are not sure if a list item applies to your research, read the appropriate section before selecting a response.

## Materials & experimental systems

| n/a | Involved in the study |
|---|---|
| ☒ ☐ | Antibodies |
| ☒ ☐ | Eukaryotic cell lines |
| ☒ ☐ | Palaeontology and archaeology |
| ☒ ☐ | Animals and other organisms |
| ☒ ☐ | Clinical data |
| ☒ ☐ | Dual use research of concern |
| ☒ ☐ | Plants |

## Methods

| n/a | Involved in the study |
|---|---|
| ☒ ☐ | ChIP-seq |
| ☒ ☐ | Flow cytometry |
| ☒ ☐ | MRI-based neuroimaging |

## Plants

| | |
|---|---|
| Seed stocks | n/a (no plants invovled) |
| Novel plant genotypes | n/a |
| Authentication | n/a |

