## [Peer Review file · Nature Microbiology]

An updated evolutionary classification of CRISPR-Cas systems including rare variants

Corresponding Author: Dr Eugene Koonin

Version 0:

Reviewer comments:

Reviewer #1

(Remarks to the Author)

Dear editor, dear authors,

Key results

In this work, building on previous classifications, Makarova and colleagues provide an updated classification of CRISPR-Cas systems and their accessory proteins. This update includes surprising insights into the distribution of CRISPR systems within newly available archaeal genome sequences, suggesting that CRISPR-Cas is not as prevalent in archaea as previously thought. Furthermore, the authors place the newly discovered type VII CRISPR system in context with other class 1 systems and describe newly identified effector domains recently found associated with class 1 system subtypes.

Within this description of class 1 systems, the authors provide an updated classification of accessory effector proteins of the type III systems, which are activated by small cyclic oligonucleotide secondary messengers. Such a classification system is needed in an expanding field with rapidly expanding numbers of accessory proteins, activator molecules, and immune mechanisms. Furthermore, the authors provide an update on new class 2 subtypes and variants discovered in recent years, including systems evolving away from traditional immune system roles to transcriptional regulation and guided transposition. Special attention is given to new insights into the evolution of types II, V and VI systems revealed by recent works. Additionally, these evolutionary insights are extensively discussed to propose mechanisms describing the emergence of diverse CRISPR systems.

Validity

I am not an expert in the bioinformatic pipelines used to generate many of the conclusions in this manuscript. However, to the best of my knowledge, most conclusions are consistent with other work in the field (although sometimes descriptions can be phrased more accurately, see below).

Significance

The findings presented in this manuscript represent an important and necessary update to CRISPR-Cas biology, evolution and classification. They will be of interest to readers of Nature reviews in microbiology who are interested in CRISPR system biology or evolution or bacterial immunity in generally. Previous (now outdated) versions of similar reviews by the same authors have been cited thousands of times, highlighting their importance and scope.

Data and methodology

As mentioned above, I cannot address the bioinformatic methodology used here. However, there do appear to be some minor errors in the presentation of the data, see below.

Suggested improvements

In its current form, the description of the reclassification of type III accessory proteins is difficult to follow and is not presented in a way that allows for easy classification of newly discovered accessory proteins. It appears that the proposed new classification is based on phylogeny, is that correct? Does classification also depend on activating molecule, on the effector domain, on the presence of ring nuclease activity, etc.? The application of legacy names without accompanying citations within figures is also confusing.

I suggest organizing the new classification within a table in the main text with new protein classification names shown with legacy names, CARF/SAVED domain, activating molecule, known and predicted activities, corresponding citations, etc, for completeness. Presenting data in this way could help the reader clearly understand what sets the different type III effector proteins apart from each other and would make the proposed reclassification more impactful/useful.

Clarity and context

The text meanders in places and repeated references to important points discussed later or earlier in the text suggest that sections might be reordered to clarify the presented information.

Many sentences are unnecessarily lengthy (e.g.: line 131...and for type VI, evidence has been obtained that shows they originate from AbiF, a type III toxin-antitoxin (TA) module. > and type VI systems originate from AbiF, a type III toxin-antitoxin (TA) module) or unclear.

I feel that certain sections of the manuscript would benefit from further proofreading to enhance readability. I have made various comments about textual unclarity and suggestions for improvements below.

Possibly, it might help to add additional headers (if the journal allows) to better organize data and make it more clear what information can be found where.

References

At various places key references are missing. I have made several suggestions on where to include references below. As this is a key resource manuscript, which many people will use to find other work, it will be crucial to include all relevant references.

Expertise

My expertise is in protein biochemistry and structural biology, particularly in CRISPR-Cas systems. I am not qualified to comment on bioinformatic pipelines used here.

Minor comments

1. This might be a personal preference, but I would suggest to use 'CRISPR-Cas' instead of 'CRISPR'. It is only one more word but is more accurate (i.e. the Cas proteins are important for the system) and prevents confusion with CRISPR arrays/loci. Furthermore, Cas proteins are what is used here to determine the classification, so it would make sense to include. Of note, also in their previous reviews, the authors have used CRISPR-Cas instead of CRISPR, so I do not see why it is necessary to break with this tradition.

2. The authors should consider condensing the three paragraphs describing the background on CRISPR-Cas system biology (lines 80-99) into a few sentences.

a. The authors admit that the stages of CRISPR system biology have been extensively reviewed elsewhere.

b. They are not particularly relevant for understanding the scope of this paper (unless a requirement of this journal). A description of the stages of CRISPR immunity should focus on functional gene modules and not on the biological function.

c. In attempting to be brief/simplified, the authors have made their descriptions unclear or incorrect. For example, the description of adaptation on lines 82-84 is incorrect:

i. "CRISPR-associated (Cas) proteins binds to a target DNA":

1. The authors should clarify that Cas1, 2, and/or 4 are the Cas proteins described here. Other Cas proteins may be required in different systems, but these represent the core adaptation module.

2. Usually protospacers are selected from a pool of short DNA fragments generated by RecBCD or Cas3.

3. Using "target" implies crRNA dependent targeting for adaptation which can occur in primed adaptation, but is by no means the rule.

ii. "... cleaves out a portion of the target DNA complementary to the spacer, the protospacer.":

1. The portion of DNA cleaved and processed by the Cas1:Cas2 complex does not need to be complementary to a spacer (i.e. there might not be prior knowledge of the protospacer in the form of a spacer).

2. Protospacer describes a sequence that is destined to be integrated into the CRISPR array. For clarity, for this reason "protospacer" in line 96 should be changed to target DNA.

3. While crRNA-dependent DNA targeting may lead to primed adaptation, this is not necessarily adaptation of a sequence that is complementary to the existing crRNA spacer.

3. In several places (e.g. lines, 160, 223, 277-278, 349, 432, 521-524, figure 5) the authors say target when cleavage is meant. This distinction is important because not all cleavage substrates are targeted by crRNA base-pairing. This distinction is made well in descriptions of type V systems (lines 399 – 408).

a. Line 246: Do the authors maybe mean DNA targets?

4. As mentioned in the suggested improvements, the current form of type III effector reclassification is not entirely clear, and it could be made more. For example, Csm6 from *T. onnurineus* is activated by cA4 (Jia et al., Mol Cell 2019) while, Csm6 from *S. thermophilus* is activated by cA6 (Smalakyte et al., NAR 2020). It is unclear from the reclassification here if these reported Csm6 proteins are both classified as Crf1, if they are classified as different subtypes of Crf1 and how these particular systems relate to other characterized and uncharacterized systems. A reader cannot take the information provided here and classify their own, newly discovered type III accessory nuclease, which might be very useful in this rapidly expanding field.

5. Feedback on figures

a. Figure 1:

i. B: What is meant with weight fraction of genomes? Perhaps more clear to indicate as: Fraction of genomes in which specific CRISPR-Cas types are found.

ii. C: In its current form, the heat map does not provide much more information than 1B. Because the scale between Type I and Type IV makes anything other than Types I and III appear as if there are no CRISPR systems in these phyla. Maybe, dividing heat map into maps by different type to reveal the actual distribution of types other than I and III.

iii. D,E: Are these figures meaningful additions to this figure? D and E are essentially the same. Basically this highlights that the current CRISPR field remains much the same as it was in the last CRISPR update.

iv. D: It would be helpful to include a column with the old and new subtypes combined to provide an accurate abundance of the total subtypes.

b. Figure 2:

i. B: Could it be indicated what the line in panel B represents (in the figure itself) for clarity)?

c. Figure 3:

i. B: The construction of this panel appears sloppy. E.g. corners of line segments should be joined for consistent corners.

ii. B: The speech bubbles are messy and are not clear. Parallelism is also missing in descriptions.

iii. B: Speech bubble for membrane proteins is much bigger than necessary. If gradients are applied to speech bubbles it should be applied to all.

iv. A: While a legend for the different colored clades is provided, it was not directly clear to me what the colored markings around the phylogenetic tree indicate (although I later noticed it is indicated in the rings. Perhaps a legend would be more clear, also for readability (i.e. font size is really small). Furthermore, perhaps it would be insightful to match the colors in Panel A and Panel B somehow.

v. B: oligoA abbreviate to cOA for consistency?

d. Figure 4:

i. C and D: Cleaves column has incorrect descriptions, for example, Cas12g cleaves RNA AND ssDNA while Cas12a2 cleaves ss, dsDNA AND RNA. This section may also benefit from adding a column describing targeting (i.e. the type of nucleotide bound by the crRNA) which is also more nuanced than what is being cleaved.

ii. E: Type V-A2 listed as containing tracrRNA (but this is not described in Dmytrenko et al., 2023) and not described anywhere else in the text?

e. Figure 5:

i. As for figure 3B, this figure appears sloppy at places. E.g. corners of line segments should be joined for consistent corners.

ii. Title of the green box (transposable elements) should be placed on top for consistency.

f. Sup. Fig 6. Low resolution structure images, not publication quality.

g. Sup. Fig 8. Crf11 clade is invisible? Suggest to change color.

Further minor textual comments and typos (numbers indicate lines):

108: of these remarkable systems

116: this suggests that the current manuscript is going to use another approach.

128: ones  systems

143:  and Cas2, which forms

144: which subtypes? Refs missing.

147: ref missing.

148: ref missing.

155: it should be made clear here that target DNA (or RNA) recognition occurs through utilization of a crRNA.

158: refs missing

161: refs missing

164: refs missing

167: refs missing

168: yes, but in addition also that they generally generate single DNA cuts at well-defined places.

169: The ancillary module of what? This sentence is a bit unclear and appears to come out of nowhere.

194: "New variants that were identified in recent genome and metagenome analyses that were absent in the complete genome collection were added to the list of CRISPR loci (Supplementary Figures 1-4, Supplementary Dataset 4)."  isn't this mentioned before in line 186? Or what is the difference?

176/177/184/197: what kind of profiles? HMM Profiles? PSI-BLAST profiles? For clarity better to be complete.

206: Ref? By who?

208: of subtypes in this class of type I subtypes

213-214: perhaps the authors can also mention the role of Cas7 and Cas5 for clarity.

223: target and cleave instead of target (i.e. nuclease activity is not required for targeting).

246: can be tentatively predicted are predicted

248: most intriguing This is quite arbitrary.

278: Another new variant Another new CRISPR-Cas system subtype

285: three additional (prevents double use of more in the same sentence)

315: I think it is important to indicate here that there are many distinct effector proteins.

323:  the known cOA sensors? Or is it evident that there are no other sensors?

325: unrelated what? CARF domains? Or ring nucleases?

328: refs missing

371: Please rephrase this sentence requires rephrasing for clarity

633-636: How do the authors envision this? Currently it feels a bit like a negative note to end on, while the authors have done great work in providing a good way for CRISPR-Cas classification. I feel that, due to shuffling and divergence, there is no 'best' way to do a classification (even if we scientists like to accurately classify everything).

Reviewer #2

(Remarks to the Author)

The current manuscript by Makarov et al 2025 is a comprehensive compendium of prokaryotic CRISPR systems and their classifications. It is well written, has a clear purpose and addresses the stated aims. This manuscript will undoubtedly be highly cited and be of significant interest to those involved in CRISPR research, both basic and applied. It is an extremely dense and

detailed review. This does create a high barrier to entry when reading that requires a significant understanding of CRISPR biology. This is not to negate its significance but may limit interest outside of the immediate field.

I have some general criticisms regarding the preparation of the manuscript in its current state. Most figures are poorly presented, which detracts from the quality and clear impact of this manuscript. There is inconsistent graph sizing, font sizing and colour selection. In many instances, it is difficult to pair what is being mentioned in the text to what the figures are presenting. Many of the figures are also devoted to the variation in CRISPR genomic organisation, whilst the later sections on distribution and evolution, which I found to be extremely informative and captivating, lack suitable figures (other than figure 5 which has its pluses but is still largely left wanting).

The sheer level of variation within subtypes, in particular Cas12, is daunting. Whilst this is extremely interesting does every variant within a subtype need distinct classification, or would a generalised principle be enough for classification (i.e. box1) with the acknowledgement that variation exists within species. For example, V-F04 to V-F13, is there need for these to be subdivided into distinct variant classes. What value does this distinction add. This is not a criticism, but merely an alternative thought for the way in which these systems are classified.

- Line 100: I would remove the word obviously. Tone seems off when it is used.
- Figure 1 requires improvement. There is big variation in text size and colour palettes. Why are D and E so large compared to A and B
- Figure 2 is hard to follow. There is a lot going on with regard to the different subtypes.
- The novelties with regard to genomic organisation for each crispr subtype are highlighted in the figure, yet variations in function(method of targeting) are only mentioned in the text. It would be helpful as a reader to have the functional consequences of this genomic divergence displayed in the same figure. Maybe a simple subheading above each subfigure is all it needs.
- Font on figure three is extremely small in places
- Figure 4 requires significant improvement. I like the general idea behind 4C and D but why is it only used here and not for other types. 4C and D are also poorly presented.
- The distribution of CRISPR systems in bacteria and archaea is a really interesting section. Figure 1C though could use improvement. Maybe just some gridlines to improve clarity. Also there is no scale bar for the heatmap.
- Figure 5 is great, and has a lot of potential to be a really important part of this review. It's a shame that the text doesn't line with the figure all that well. I think some expansion of figure 5 to highlight how different subtypes/variants are being assembled would be beneficial and make the review more accessible. Figure 5 also needs a legend.
- There is large number of authors associated with this manuscript. All are significant players in CRISPR biology. Whilst this does give weight and support to this system of classification within the field, is the inclusion of all authors necessary? K.S.M., S.A.S., Y.I.W., P.M., P.P. and E.V.K. did the research and wrote the article, is this not sufficient for the authorship. I think some further justification for the number of researchers should be included.

Reviewer #3

(Remarks to the Author)

Decision Letter:

18th June 2025

Dear Eugene,

Thank you for your patience while your manuscript "Classification and Evolution of CRISPR systems: Discoveries in the Tail of the Distribution" was under peer-review at Nature Microbiology. So sorry for the delay! We had one reviewer that gave us the runaround... Your paper has now been seen by 2 referees, whose expertise and comments you will find at the end of this email. Although they find your work of some potential interest, and agree that this will be an important contribution to the field (which we agree about!), they have raised a number of concerns that will need to be addressed before we can consider publication of the work in Nature Microbiology.

In particular, the biggest issue raised by both referees is with regards to the clarity and presentation of the text. They agree that the work would benefit from a bit of streamlining and editing with a more general microbiology audience in mind. While the topic is necessarily dense and detailed, it should be as accessible as possible. They also had a helpful suggestion to include organizing the new classification within a table in the main text, which we agree is a good idea.

Should further experimental data allow you to address these criticisms, we would be happy to look at a revised manuscript.

Please include a data availability statement as a separate section after Methods but before references, under the heading "Data Availability". This section should inform readers about the availability of the data used to support the conclusions of your study. This information includes accession codes to public repositories (data banks for protein, DNA or RNA sequences, microarray, proteomics data etc...), references to source data published alongside the paper, unique identifiers such as URLs to data repository entries, or data set DOIs, and any other statement about data availability. At a minimum, you should include the following statement: "The data that support the findings of this study are available from the corresponding author upon request", mentioning any restrictions on availability. If DOIs are provided, we also strongly encourage including these in the Reference list (authors, title, publisher (repository name), identifier, year). For more guidance on how to write this section please see: <http://www.nature.com/authors/policies/data/data-availability-statements-data-citations.pdf>

* If you have not done so already we suggest that you begin to revise your manuscript so that it conforms to our Analysis format instructions at <http://www.nature.com/nmicrobiol/info/final-submission>. Refer also to any guidelines provided in this letter.

When submitting the revised version of your manuscript, please pay close attention to our [href="https://www.nature.com/nature-portfolio/editorial-policies/image-integrity">Digital Image Integrity Guidelines.](https://www.nature.com/nature-portfolio/editorial-policies/image-integrity) and to the following points below:

EXTENDED DATA FIGURES

Link Redacted

Note: This url links to your confidential homepage and associated information about manuscripts you may have submitted or be reviewing for us. If you wish to forward this e-mail to co-authors, please delete this link to your homepage first.

Nature Microbiology is committed to improving transparency in authorship. As part of our efforts in this direction, we are now requesting that all authors identified as 'corresponding author' on published papers create and link their Open Researcher and Contributor Identifier (ORCID) with their account on the Manuscript Tracking System (MTS), prior to acceptance. This applies to primary research papers only. ORCID helps the scientific community achieve unambiguous attribution of all scholarly contributions. You can create and link your ORCID from the home page of the MTS by clicking on 'Modify my Springer Nature account'. For more information please visit www.springernature.com/orcid.

If you wish to submit a suitably revised manuscript we would hope to receive it within 6 months. If you cannot send it within this time, please let us know. We will be happy to consider your revision, even if a similar study has been accepted for publication at Nature Microbiology or published elsewhere (up to a maximum of 6 months).

Yours sincerely,

Reviewer Expertise:

Referee #1: phage defense systems, CRISPR, structural biology

Referee #2: CRISPR, genetics, functional genomics

Reviewer Comments:

Reviewer #1 (Remarks to the Author):

Dear editor, dear authors,

Key results

In this work, building on previous classifications, Makarova and colleagues provide an updated classification of CRISPR-Cas systems and their accessory proteins. This update includes surprising insights into the distribution of CRISPR systems within newly available archaeal genome sequences, suggesting that CRISPR-Cas is not as prevalent in archaea as previously thought. Furthermore, the authors place the newly discovered type VII CRISPR system in context with other class 1 systems and describe newly identified effector domains recently found associated with class 1 system subtypes.

Within this description of class 1 systems, the authors provide an updated classification of accessory effector proteins of the type III systems, which are activated by small cyclic oligonucleotide secondary messengers. Such a classification system is needed in an expanding field with rapidly expanding numbers of accessory proteins, activator molecules, and immune mechanisms. Furthermore, the authors provide an update on new class 2 subtypes and variants discovered in recent years, including systems evolving away from traditional immune system roles to transcriptional regulation and guided transposition. Special attention is given to new insights into the evolution of types II, V and VI systems revealed by recent works. Additionally, these evolutionary insights are extensively discussed to propose mechanisms describing the emergence of diverse CRISPR systems.

Validity

I am not an expert in the bioinformatic pipelines used to generate many of the conclusions in this manuscript. However, to the best of my knowledge, most conclusions are consistent with other work in the field (although sometimes descriptions can be phrased more accurately, see below).

Significance

The findings presented in this manuscript represent an important and necessary update to CRISPR-Cas biology, evolution and classification. They will be of interest to readers of Nature reviews in microbiology who are interested in CRISPR system biology or evolution or bacterial immunity in generally. Previous (now outdated) versions of similar reviews by the same authors have been cited thousands of times, highlighting their importance and scope.

Data and methodology

As mentioned above, I cannot address the bioinformatic methodology used here. However, there do appear to be some minor errors in the presentation of the data, see below.

Suggested improvements

In its current form, the description of the reclassification of type III accessory proteins is difficult to follow and is not presented in a way that allows for easy classification of newly discovered accessory proteins. It appears that the proposed new classification is based on phylogeny, is that correct? Does classification also depend on activating molecule, on the effector domain, on the presence of ring nuclease activity, etc.? The application of legacy names without accompanying citations within figures is also confusing.

I suggest organizing the new classification within a table in the main text with new protein classification names shown with legacy names, CARF/SAVED domain, activating molecule, known and predicted activities, corresponding citations, etc, for completeness. Presenting data in this way could help the reader clearly understand what sets the different type III effector proteins apart from each other and would make the proposed reclassification more impactful/useful.

Clarity and context

The text meanders in places and repeated references to important points discussed later or earlier in the text suggest that sections might be reordered to clarify the presented information.

Many sentences are unnecessarily lengthy (e.g.: line 131...and for type VI, evidence has been obtained that shows they originate from AbiF, a type III toxin-antitoxin (TA) module. > and type VI systems originate from AbiF, a type III toxin-antitoxin (TA) module) or unclear.

I feel that certain sections of the manuscript would benefit from further proofreading to enhance readability. I have made various

comments about textual unclarities and suggestions for improvements below.

Possibly, it might help to add additional headers (if the journal allows) to better organize data and make it more clear what information can be found where.

References

At various places key references are missing. I have made several suggests on where to include references below. As this is a key resource manuscript, which will many people will use to find other work, it will be crucial to include all relevant references.

Expertise

My expertise in in protein biochemistry and structural biology, particularly in CRISPR-Cas systems. I am not qualified to comment on bioinformatic pipelines used here.

Minor comments

1. This might be a personal preference, but I would suggest to use 'CRISPR-Cas' instead of 'CRISPR'. It is only one more word but is more accurate (i.e. the Cas proteins are important for the system) and prevents confusion with CRISPR arrays/loci. Furthermore, Cas proteins are what is used here to determine the classification, so it would make sense to include. Of note, also in their previous reviews, the author have used CRISPR-Cas instead of CRISPR, so I do not see why it is necessary to break with this tradition.

2. The authors should consider condensing the three paragraphs describing the background on CRISPR-Cas system biology (lines 80-99) into a few sentences.

a. The authors admit that the stages of CRISPR system biology have been extensively reviewed elsewhere.

b. They are not particularly relevant for understanding the scope of this paper (unless a requirement of this journal). A description of the stages of CRISPR immunity should focus on functional gene modules and not on the biological function.

c. In attempting to be brief/simplified, the authors have made their descriptions unclear or incorrect. For example, the description of adaptation on lines 82-84 is incorrect:

i. "CRISPR-associated (Cas) proteins binds to a target DNA":

1. The authors should clarify that Cas1, 2, and/or 4 are the Cas proteins described here. Other Cas proteins may be required in different systems, but these represent the core adaptation module.

2. Usually protospacers are selected from a pool of short DNA fragments generated by RecBCD or Cas3.

3. Using "target" implies crRNA dependent targeting for adaptation which can occur in primed adaptation, but is by no means the rule.

ii. "... cleaves out a portion of the target DNA complementary to the spacer, the protospacer.":

1. The portion of DNA cleaved and processed by the Cas1:Cas2 complex does not need to be complimentary to a spacer (i.e. there might not be prior knowledge of the protospacer in the form of a spacer).

2. Protospacer describes a sequence that it destined to be integrated into the CRISPR array. For clarity, for this reason "protospacer" in line 96 should be changed to target DNA.

3. While crRNA-dependent DNA targeting may lead to primed adaptation, this is not necessarily adaptation of a sequence that is complementary to the existing crRNA spacer.

3. In several places (e.g. lines, 160, 223, 277-278, 349, 432, 521-524, figure 5) the authors say target when cleavage is meant. This distinction is important because not all cleavage substrates are targeted by crRNA base-pairing. This distinction is made well in descriptions of type V systems (lines 399 – 408).

a. Line 246: Do the authors maybe mean DNA targets?

4. As mentioned in the suggested improvements, the current form of type III effector reclassification is not entirely clear, and it could be made more. For example, Csm6 from *T. onnurineus* is activated by cA4 (Jia et al., Mol Cell 2019) while, Csm6 from *S. thermophilus* is activated by cA6 (Smalakyte et al., NAR 2020). It is unclear from the reclassification here if these reported Csm6 proteins are both classified as Crf1, if they are classified as different subtypes of Crf1 and how these particular systems relate to other characterized and uncharacterized systems. A reader cannot take the information provided here and classify their own, newly discovered type III accessory nuclease, which might be very useful in this rapidly expanding field.

5. Feedback on figures

a. Figure 1:

i. B: What is meant with weight fraction of genomes? Perhaps more clear to indicate as: Fraction of genomes in which specific CRISPR-Cas types are found.

ii. C: In its current form, the heat map does not provide much more information than 1B. Because the scale between Type I and Type IV makes anything other than Types I and III appear as if there are no CRISPR systems in these phyla. Maybe, dividing heat map into maps by different type to reveal the actual distribution of types other than I and III.

iii. D,E: Are these figures meaningful additions to this figure? D and E are essentially the same. Basically this highlights that the current CRISPR field remains much the same as it was in the last CRISPR update.

iv. D: It would be helpful to include a column with the old and new subtypes combined to provide an accurate abundance of the total subtypes.

b. Figure 2:

i. B: Could it be indicated what the line in panel B represents (in the figure itself) for clarity)?

c. Figure 3:

i. B: The construction of this panel appears sloppy. E.g. corners of line segments should be joined for consistent corners.

ii. B: The speech bubbles are messy and are not clear. Parallelism is also missing in descriptions.

iii. B: Speech bubble for membrane proteins is much bigger than necessary. If gradients are applied to speech bubbles it should

be applied to all.

iv. A: While a legend for the different colored clades is provided, it was not directly clear to me what the colored markings around the phylogenetic tree indicate (although I later noticed it is indicated in the rings. Perhaps a legend would be more clear, also for readability (i.e. font size is really small). Furthermore, perhaps it would be insightful to match the colors in Panel A and Panel B somehow.

v. B: oligoA abbreviate to cOA for consistency?

d. Figure 4:

i. C and D: Cleaves column has incorrect descriptions, for example, Cas12g cleaves RNA AND ssDNA while Cas12a2 cleaves ss, dsDNA AND RNA. This section may also benefit from adding a column describing targeting (i.e. the type of nucleotide bound by the crRNA) which is also more nuanced than what is being cleaved.

ii. E: Type V-A2 listed as containing tracrRNA (but this is not described in Dmytrenko et al., 2023) and not described anywhere else in the text?

e. Figure 5:

i. As for figure 3B, this figure appears sloppy at places. E.g. corners of line segments should be joined for consistent corners.

ii. Title of the green box (transposable elements) should be placed on top for consistency.

f. Sup. Fig 6. Low resolution structure images, not publication quality.

g. Sup. Fig 8. Crf11 clade is invisible? Suggest to change color.

Further minor textual comments and typos (numbers indicate lines):

108: of these remarkable systems

116: this suggests that the current manuscript is going to use another approach.

128: ones  systems

143:  and Cas2, which forms

144: which subtypes? Refs missing.

147: ref missing.

148: ref missing.

155: it should be made clear here that target DNA (or RNA) recognition occurs through utilization of a crRNA.

158: refs missing

161: refs missing

164: refs missing

167: refs missing

168: yes, but in addition also that they generally generate single DNA cuts at well-defined places.

169: The ancillary module of what? This sentence is a bit unclear and appears to come out of nowhere.

194: "New variants that were identified in recent genome and metagenome analyses that were absent in the complete genome collection were added to the list of CRISPR loci (Supplementary Figures 1-4, Supplementary Dataset 4)." -> isn't this mentioned before in line 186? Or what is the difference?

176/177/184/197: what kind of profiles? HMM Profiles? PSI-BLAST profiles? For clarity better to be complete.

206: Ref? By who?

208: of subtypes in this class of type I subtypes

213-214: perhaps the authors can also mention the role of Cas7 and Cas5 for clarity.

223: target and cleave instead of target (i.e. nuclease activity is not required for targeting).

246: can be tentatively predicted are predicted

248: most intriguing This is quite arbitrary.

278: Another new variant Another new CRISPR-Cas system subtype

285: three additional (prevents double use of more in the same sentence)

315: I think it is important to indicate here that there are many distinct effector proteins.

323:  the known cOA sensors? Or is it evident that there are no other sensors?

325: unrelated what? CARF domains? Or ring nucleases?

328: refs missing

371: Please rephrase this sentence requires rephrasing for clarity

633-636: How do the authors envision this? Currently it feels a bit like a negative note to end on, while the authors have done great work in providing a good way for CRISPR-Cas classification. I feel that, due to shuffling and divergence, there is no 'best' way to do a classification (even if we scientists like to accurately classify everything).

Reviewer #2 (Remarks to the Author):

The current manuscript by Makarov et al 2025 is a comprehensive compendium of prokaryotic CRISPR systems and their classifications. It is well written, has a clear purpose and addresses the stated aims. This manuscript will undoubtedly be highly cited and be of significant interest to those involved in CRISPR research, both basic and applied. It is an extremely dense and detailed review. This does create a high barrier to entry when reading that requires a significant understanding of CRISPR biology. This is not to negate its significance but may limit interest outside of the immediate field.

I have some general criticisms regarding the preparation of the manuscript in its current state. Most figures are poorly presented, which detracts from the quality and clear impact of this manuscript. There is inconsistent graph sizing, font sizing and colour selection. In many instances, it is difficult to pair what is being mentioned in the text to what the figures are presenting. Many of the figures are also devoted to the variation in CRISPR genomic organisation, whilst the later sections on distribution and

evolution, which I found to be extremely informative and captivating, lack suitable figures (other than figure 5 which has its pluses but is still largely left wanting).

The sheer level of variation within subtypes, in particular Cas12, is daunting. Whilst this is extremely interesting does every variant within a subtype need distinct classification, or would a generalised principle be enough for classification (i.e. box1) with the acknowledgement that variation exists within species. For example, V-F04 to V-F13, is there need for these to be subdivided into distinct variant classes. What value does this distinction add. This is not a criticism, but merely an alternative thought for the way in which these systems are classified.

- Line 100: I would remove the word obviously. Tone seems off when it is used.
- Figure 1 requires improvement. There is big variation in text size and colour palettes. Why are D and E so large compared to A and B
- Figure 2 is hard to follow. There is a lot going on with regard to the different subtypes.
- The novelties with regard to genomic organisation for each crispr subtype are highlighted in the figure, yet variations in function(method of targeting) are only mentioned in the text. It would be helpful as a reader to have the functional consequences of this genomic divergence displayed in the same figure. Maybe a simple subheading above each subfigure is all it needs.
- Font on figure three is extremely small in places
- Figure 4 requires significant improvement. I like the general idea behind 4C and D but why is it only used here and not for other types. 4C and D are also poorly presented.
- The distribution of CRISPR systems in bacteria and archaea is a really interesting section. Figure 1C though could use improvement. Maybe just some gridlines to improve clarity. Also there is no scale bar for the heatmap.
- Figure 5 is great, and has a lot of potential to be a really important part of this review. It's a shame that the text doesn't line with the figure all that well. I think some expansion of figure 5 to highlight how different subtypes/variants are being assembled would be beneficial and make the review more accessible. Figure 5 also needs a legend.
- There is large number of authors associated with this manuscript. All are significant players in CRISPR biology. Whilst this does give weight and support to this system of classification within the field, is the inclusion of all authors necessary? K.S.M., S.A.S., Y.I.W., P.M., P.P. and E.V.K. did the research and wrote the article, is this not sufficient for the authorship. I think some further justification for the number of researchers should be included.

Version 1:

Reviewer comments:

Reviewer #1

(Remarks to the Author)

Dear editor, dear authors,

The authors have put major efforts in increasing the correctness and completeness of information displayed, citations, and readability/flow. The manuscript now reads a lot better than the previous version, and the author have made clear efforts in clarifying the classification of or type III accessory proteins.

Table 1 aims at providing the much needed clarity about the type III effector classification. While this is a good start, there are various unclarities in this table. Crf7 is listed in both the cOA sensors-effectors and ring nucleases, but the old names and superfamily descriptions are not the same. It seems like the superfamily descriptions should be the same for the same proteins. Crf9 old names are also inconsistent between these two sections of the table. Several proteins in the sensor-effector portion of the table are described as having (or possibly having) ring nuclease activity, but are not included in the ring nuclease section of the table (eg. Crf1 and 9, and CalpL). Furthermore, Crf5 is described as having effector and ring nuclease function in the ring nuclease section but is not included in the sensor-effector section of the table. As is evident from this comment, the Table still needs work to provide further clarity for the classification.

I still feel the manuscript meanders here and there, and I feel that the flow can be improved in order to help the reader comprehend the interesting information more easily. While this might be personal preference, I have given some more directions that can improve flow:

-Cluster insights into class 1 type I, III, and IV systems (in that order) in the first section

-First explain new insights into class 1 (I, III, IV) and class 2 (II and V, VI) systems, (and HRAMP/ARAMP), only then move on to distribution, and finally to type III effector diversity and system evolution.

-Additional subheadings might help readers (if allowed)

-It might help the reader to further refer to Box 1 and figures at multiple instances (e.g. box 1 is only referred to once while explaining the different system subunit compositions), while it is relevant for many other statements made. The same is true for statements in which it would help to refer to figures. Also Figure numbering is in the wrong order in the text - this will require careful proofreading. u

I hope that the authors find my comments useful and that they will help to improve their manuscript.

Reviewer #2

(Remarks to the Author)

The authors have responded to my prior comments. They have also provided suitable comments to other reviewers.

Decision Letter:

Our ref: NMICROBIOL-25041168A

26th August 2025

Dear Eugene,

Thank you for submitting your revised manuscript "Classification and Evolution of CRISPR systems: Discoveries in the Tail of the Distribution" (NMICROBIOL-25041168A). It has now been seen by the original referees and their comments are below. The reviewers find that the paper has improved in revision, and therefore we'll be happy in principle to publish it in Nature Microbiology, pending minor revisions to satisfy the referees' final requests and to comply with our editorial and formatting guidelines.

Thank you again for your interest in Nature Microbiology Please do not hesitate to contact me if you have any questions.

Sincerely,

Reviewer #1 (Remarks to the Author):

Dear editor, dear authors,

The authors have put major efforts in increasing the correctness and completeness of information displayed, citations, and readability/flow. The manuscript now reads a lot better than the previous version, and the author have made clear efforts in clarifying the classification of or type III accessory proteins.

Table 1 aims at providing the much needed clarity about the type III effector classification. While this is a good start, there are various unclaritys in this table. Crf7 is listed in both the cOA sensors-efectors and ring nucleases, but the old names and superfamily descriptions are not the same. It seems like the superfamily descriptions should be the same for the same proteins. Crf9 old names are also inconsistent between these two sections of the table. Several proteins in the sensor-effector portion of the table are described as having (or possibly having) ring nuclease activity, but are not included in the ring nuclease section of the table (eg. Crf1 and 9, and CalpL). Furthermore, Crf5 is described as having effector and ring nuclease function in the ring nuclease section but is not included in the sensor-effector section of the table. As is evident from this comment, the Table still needs work to provide further clarity for the classification.

I still feel the manuscript meanders here and there, and I feel that the flow can be improved in order to help the reader comprehend the interesting information more easily. While this might be personal preference, I have given some more directions that can improve flow:

-Cluster insights into class 1 type I, III, and IV systems (in that order) in the first section

-First explain new insights into class 1 (I, III, IV) and class 2 (II and V, VI) systems, (and HRAMP/ARAMP), only then move on to distribution, and finally to type III effector diversity and system evolution.

-Additional subheadings might help readers (if allowed)

-It might help the reader to further refer to Box 1 and figures at multiple instances (e.g. box 1 is only referred to once while explaining the different system subunit compositions), while it is relevant for many other statements made. The same is true for statements in which it would help to refer to figures. Also Figure numbering is in the wrong order in the text - this will require careful proofreading. u

I hope that the authors find my comments useful and that they will help to improve their manuscript.

Reviewer #2 (Remarks to the Author):

The authors have responded to my prior comments. They have also provided suitable comments to other reviewers.

Version 2:

Decision Letter:

7th October 2025

Dear Eugene,

I am pleased to accept your Analysis "An updated evolutionary classification of CRISPR-Cas systems including rare variants" for publication in Nature Microbiology. Thank you for having chosen to submit your work to us and many congratulations.

Authors may need to take specific actions to achieve compliance with funder and institutional open access mandates. If your research is supported by a funder that requires immediate open access (e.g. according to [a href="https://www.springernature.com/gp/open-science/plan-s-compliance"> Plan S principles](https://www.springernature.com/gp/open-science/plan-s-compliance) or the [a href="https://www.springernature.com/gp/open-science/us-federal-agency-compliance"> NIH public access policy](https://www.springernature.com/gp/open-science/us-federal-agency-compliance)) then you should select the gold OA route, and we will direct you to the compliant route where possible. Because authors warrant under our subscription licensing terms that they haven't committed to licensing any version of their article under a licence inconsistent with the terms of our agreement – including the applicable embargo period – publication under the subscription model isn't suitable for authors whose funders require no embargo.

An online order form for reprints of your paper is available at [a href="https://www.nature.com/reprints/author-reprints.html">https://www.nature.com/reprints/author-reprints.html](https://www.nature.com/reprints/author-reprints.html). All co-authors, authors' institutions and authors' funding agencies can order reprints using the form appropriate to their geographical region.

We welcome the submission of potential cover material (including a short caption of around 40 words) related to your manuscript; suggestions should be sent to Nature Microbiology as electronic files (the image should be 300 dpi at 210 x 297 mm in either TIFF or JPEG format). Please note that such pictures should be selected more for their aesthetic appeal than for their

scientific content, and that colour images work better than black and white or grayscale images. Please do not try to design a cover with the Nature Microbiology logo etc., and please do not submit composites of images related to your work. I am sure you will understand that we cannot make any promise as to whether any of your suggestions might be selected for the cover of the journal.

With kind regards,

P.S. Click on the following link if you would like to recommend Nature Microbiology to your librarian
<http://www.nature.com/subscriptions/recommend.html#forms>

** Visit the Springer Nature Editorial and Publishing website at http://editorial-jobs.springernature.com?utm_source=ejP_NMicro_email&utm_medium=ejP_NMicro_email&utm_campaign=ejp_NMicro for more information about our career opportunities. If you have any questions please click [here](mailto:editorial.publishing.jobs@springernature.com).**

Reviewer #1 (Remarks to the Author):

Dear editor, dear authors,

Key results

In this work, building on previous classifications, Makarova and colleagues provide an updated classification of CRISPR-Cas systems and their accessory proteins. This update includes surprising insights into the distribution of CRISPR systems within newly available archaeal genome sequences, suggesting that CRISPR-Cas is not as prevalent in archaea as previously thought. Furthermore, the authors place the newly discovered type VII CRISPR system in context with other class 1 systems and describe newly identified effector domains recently found associated with class 1 system subtypes.

Within this description of class 1 systems, the authors provide an updated classification of accessory effector proteins of the type III systems, which are activated by small cyclic oligonucleotide secondary messengers. Such a classification system is needed in an expanding field with rapidly expanding numbers of accessory proteins, activator molecules, and immune mechanisms.

Furthermore, the authors provide an update on new class 2 subtypes and variants discovered in recent years, including systems evolving away from traditional immune system roles to transcriptional regulation and guided transposition. Special attention is given to new insights into the evolution of types II, V and VI systems revealed by recent works. Additionally, these evolutionary insights are extensively discussed to propose mechanisms describing the emergence of diverse CRISPR systems.

Validity

I am not an expert in the bioinformatic pipelines used to generate many of the conclusions in this manuscript. However, to the best of my knowledge, most conclusions are consistent with other work in the field (although sometimes descriptions can be phrased more accurately, see below).

Significance

The findings presented in this manuscript represent an important and necessary update to CRISPR-Cas biology, evolution and classification. They will be of interest to readers of Nature reviews in microbiology who are interested in CRISPR system biology or evolution or bacterial immunity in generally. Previous (now outdated) versions of similar reviews by the same authors have been cited thousands of times, highlighting their importance and scope.

Data and methodology

As mentioned above, I cannot address the bioinformatic methodology used here. However, there do appear to be some minor errors in the presentation of the data, see below.

We appreciate the reviewer's interest in and close attention to our work as well as the constructive, specific comments which we address below.

Suggested improvements

In its current form, the description of the reclassification of type III accessory proteins is difficult to follow and is not presented in a way that allows for easy classification of newly discovered accessory proteins. It appears that the proposed new classification is based on phylogeny, is that correct? Does

classification also depend on activating molecule, on the effector domain, on the presence of ring nuclease activity, etc.? The application of legacy names without accompanying citations within figures is also confusing.

I suggest organizing the new classification within a table in the main text with new protein classification names shown with legacy names, CARF/SAVED domain, activating molecule, known and predicted activities, corresponding citations, etc, for completeness. Presenting data in this way could help the reader clearly understand what sets the different type III effector proteins apart from each other and would make the proposed reclassification more impactful/useful.

The classification of the type III accessory proteins is indeed largely based on the phylogeny of the CARF domains inasmuch as this domain as present. We emphasize this in the revision. The table proposed by the reviewer (new Table 1) was constructed and included.

Clarity and context

The text meanders in places and repeated references to important points discussed later or earlier in the text suggest that sections might be reordered to clarify the presented information.

Many sentences are unnecessarily lengthy (e.g.: line 131...and for type VI, evidence has been obtained that shows they originate from AbiF, a type III toxin-antitoxin (TA) module. > and type VI systems originate from AbiF, a type III toxin-antitoxin (TA) module) or unclear.

I feel that certain sections of the manuscript would benefit from further proofreading to enhance readability. I have made various comments about textual unclarities and suggestions for improvements below.

Possibly, it might help to add additional headers (if the journal allows) to better organize data and make it more clear what information can be found where.

We regret that parts of the text appeared less than transparent. The entire manuscript was edited for clarity, and in particular, a number of longer sentences were split into shorter ones.

References

At various places key references are missing. I have made several suggests on where to include references below. As this is a key resource manuscript, which will many people will use to find other work, it will be crucial to include all relevant references.

Most of the references suggested by the reviewer are cited in the revision (most actually were already in the reference list, they only had to be cited in more places in the manuscript), with some exceptions pointed out below.

Expertise

My expertise in in protein biochemistry and structural biology, particularly in CRISPR-Cas systems. I am

not qualified to comment on bioinformatic pipelines used here.

Minor comments

1. This might be a personal preference, but I would suggest to use ‘CRISPR-Cas’ instead of ‘CRISPR’. It is only one more word but is more accurate (i.e. the Cas proteins are important for the system) and prevents confusion with CRISPR arrays/loci. Furthermore, Cas proteins are what is used here to determine the classification, so it would make sense to include. Of note, also in their previous reviews, the author have used CRISPR-Cas instead of CRISPR, so I do not see why it is necessary to break with this tradition.

We tried to condense the acronym: it is only one word but it is repeated hundreds of times throughout the manuscript. However, there was a debate among the coauthors on this subject, and given the opinion of the reviewer, we reverted to CRISPR-Cas.

2. The authors should consider condensing the three paragraphs describing the background on CRISPR-Cas system biology (lines 80-99) into a few sentences.

In this case, we respectfully disagree. The introduction to CRISPR biology we present is quite brief, and we believe it would be counterproductive to condense it even further, especially, given that reviewer #2 found that understanding this article “requires a significant understanding of CRISPR biology”. This said, we greatly appreciate the reviewer’s specific suggestions and corrections which have been taken into account as detailed below.

- a. The authors admit that the stages of CRISPR system biology have been extensively reviewed elsewhere.
- b. They are not particularly relevant for understanding the scope of this paper (unless a requirement of this journal). A description of the stages of CRISPR immunity should focus on functional gene modules and not on the biological function.
- c. In attempting to be brief/simplified, the authors have made their descriptions unclear or incorrect. For example, the description of adaptation on lines 82-84 is incorrect:

We sincerely regret the inaccuracy in the description of the adaptation stage. These have been corrected, with additional references cited as specified below.

i. “CRISPR-associated (Cas) proteins binds to a target DNA”:

1. The authors should clarify that Cas1, 2, and/or 4 are the Cas proteins described here. Other Cas proteins may be required in different systems, but these represent the core adaptation module.

Specified as suggested.

2. Usually protospacers are selected from a pool of short DNA fragments generated by RecBCD or Cas3.

In the case of naïve adaptation, RecBCD is responsible as indicated in the revision. We did not think we had space to introduce primed adaptation.

3. Using “target” implies crRNA dependent targeting for adaptation which can occur in primed adaptation, but is by no means the rule.

Point well taken, in the revision, we simply write ‘foreign DNA’.

ii. "... cleaves out a portion of the target DNA complementary to the spacer, the protospacer.":
1. The portion of DNA cleaved and processed by the Cas1:Cas2 complex does not need to be complimentary to a spacer (i.e. there might not be prior knowledge of the protospacer in the form of a spacer).

Corrected.

2. Protospacer describes a sequence that it destined to be integrated into the CRISPR array. For clarity, for this reason "protospacer" in line 96 should be changed to target DNA.

Yes, we agree, "protospacer" replaced with "sequence" in this case.

3. While crRNA-dependent DNA targeting may lead to primed adaptation, this is not necessarily adaptation of a sequence that is complementary to the existing crRNA spacer.

This is certainly true, the text was corrected to say "The adaptation complex binds fragments of a foreign DNA that are typically generated by the host RecBCD repair machinery, often after recognizing a short protospacer-adjacent motif (PAM)"

3. In several places (e.g. lines, 160, 223, 277-278, 349, 432, 521-524, figure 5) the authors say target when cleavage is meant. This distinction is important because not all cleavage substrates are targeted by crRNA base-pairing. This distinction is made well in descriptions of type V systems (lines 399 – 408).

a. Line 246: Do the authors maybe mean DNA targets?

We have checked all the indicated occasions and made the language more specific where ambiguity appeared possible. Yes, certainly DNA targets, this was an unfortunate typo.

4. As mentioned in the suggested improvements, the current form of type III effector reclassification is not entirely clear, and it could be made more. For example, Csm6 from *T. onnurineus* is activated by cA4 (Jia et al., Mol Cell 2019) while, Csm6 from *S. thermophilus* is activated by cA6 (Smalakyte et al., NAR 2020). It is unclear from the reclassification here if these reported Csm6 proteins are both classified as Crf1, if they are classified as different subtypes of Crf1 and how these particular systems relate to other characterized and uncharacterized systems. A reader cannot take the information provided here and classify their own, newly discovered type III accessory nuclease, which might be very useful in this rapidly expanding field.

The reason we propose reclassification of accessory proteins of type III CRISPR-Cas systems, in particular, those (the majority) containing CARF domains, is that different clades in the CARF tree included proteins with different domain compositions and/or functions but currently are given names that do not point to either evolutionary relationships or functional distinctions. The phylogeny of CARFs correlates with the signaling molecules they sense albeit less than perfectly.

In the revised manuscript, we include Table 1 that provides information on each clade of CARF proteins as well as those containing SAVED domains, including the corresponding signaling molecules. The table also includes old names, proposed new names and references. Hopefully, this clarifies the proposed classification. Given that we provide multiple alignment-based profiles for different clades of CARFs and for SAVEDs, this information should help researchers classify newly identified type III accessory proteins.

5. Feedback on figures

Figure 1:

i. B: What is meant with weight fraction of genomes? Perhaps more clear to indicate as: Fraction of genomes in which specific CRISPR-Cas types are found.

All completely sequenced genomes in the dataset were assigned relative weights based on their (approximate) phylogenetic relationships. We display the ratio of the sum of weights of genomes from a specific taxon where a particular system is present to the total sum of genome weights for this taxon, that is, the weighted fraction. We clarified it in the figure legend.

ii. C: In its current form, the heat map does not provide much more information than 1B. Because the scale between Type I and Type IV makes anything other than Types I and III appear as if there are no CRISPR systems in these phyla. Maybe, dividing heat map into maps by different type to reveal the actual distribution of types other than I and III.

Figure 1C was modified by separating the CRISPR-Cas types as well as archaea and bacteria. We hope and trust the visual appeal of the figure is much improved. For types VI and VII, the contrast remains weak but this appears unavoidable given the rarity of these systems. More details are provided in Supplementary Table 8.

iii. D,E: Are these figures meaningful additions to this figure? D and E are essentially the same. Basically this highlights that the current CRISPR field remains much the same as it was in the last CRISPR update.

Yes, we believe these are useful additions. We are indeed trying to make the point that the abundant CRISPR subtypes were already known at the last update, and new discoveries, however interesting, are “in the tail of distribution” (hence the title of this paper).

iv. D: It would be helpful to include a column with the old and new subtypes combined to provide an accurate abundance of the total subtypes.

In principle, this is a good idea. In reality, however, the new subtypes account for <1% of the total and would not be visible when combined with the old ones.

b. Figure 2:

i. B: Could it be indicated what the line in panel B represents (in the figure itself) for clarity)?

We included the following in the legend “The line schematically represents crRNA”

a. Figure 3:

i. B: The construction of this panel appears sloppy. E.g. corners of line segments should be joined for consistent corners.

Modified as suggested.

ii. B: The speech bubbles are messy and are not clear. Parallelism is also missing in descriptions.

Regrettably, the reviewer does not specifically indicate what appears messy in those bubbles. We tried to provide information that we perceived as most relevant for each class of type III components. The meaning of ‘parallelism’ is not entirely obvious. If what is meant, is uniformity of the contents of those bubbles, it is unclear how this could be achieved because the pertinent information differs.

iii. B: Speech bubble for membrane proteins is much bigger than necessary. If gradients are applied to speech bubbles it should be applied to all.

In the revision, that bubble was made smaller and the gradient was removed.

iv. A: While a legend for the different colored clades is provided, it was not directly clear to me what the colored markings around the phylogenetic tree indicate (although I later noticed it is indicated in the rings. Perhaps a legend would be more clear, also for readability (i.e. font size is really small). Furthermore, perhaps it would be insightful to match the colors in Panel A and Panel B somehow.

Colors changed to match between A and B, wherever possible.

v. B: oligoA à abbreviate to cOA for consistency?

Corrected as suggested.

d. Figure 4:

i. C and D: Cleaves column has incorrect descriptions, for example, Cas12g cleaves RNA AND ssDNA while Cas12a2 cleaves ss, dsDNA AND RNA. This section may also benefit from adding a column describing targeting (i.e. the type of nucleotide bound by the crRNA) which is also more nuanced than what is being cleaved.

The entries in the ‘Cleaves’ column were double checked and corrected where needed. We decided, however, not to introduce the proposed additional ‘Targeting’ column because these proteins have both the main, crRNA-dependent activity and additional, co-lateral activities, making things too complicated.

ii. E: Type V-A2 listed as containing tracrRNA (but this is not described in Dmytrenko et al., 2023) and not described anywhere else in the text?

We thank the reviewer for spotting this mistake. The figure was corrected. Please, note that in Figure 2C we had correct information.

e. Figure 5:

i. As for figure 3B, this figure appears sloppy at places. E.g. corners of line segments should be joined for consistent corners.

Corrected as suggested.

ii. Title of the green box (transposable elements) should be placed on top for consistency.

Modified as suggested.

f. Sup. Fig 6. Low resolution structure images, not publication quality.

Higher resolution image included.

g. Sup. Fig 8. Crf11 clade is invisible? Suggest to change color.

Color changed as suggested.

Further minor textual comments and typos (numbers indicate lines):

108: of these remarkable systems

Corrected.

116: this suggests that the current manuscript is going to use another approach.

We must respectfully disagree: there is no such implication.

128: ones  system

Corrected

143:  and Cas2, which forms

Comma inserted as suggested.

144: which subtypes? Refs missing.

We did not think it made sense to list all subtypes with Cas4 in adaptation complex, we simply indicated there are many and referred to Supplementary figures 1-3; references cited.

147: ref missing.

References cited.

148: ref missing.

References cited.

155: it should be made clear here that target DNA (or RNA) recognition occurs through utilization of a crRNA.

crRNA mentioned explicitly in the revision.

158: refs missing

References cited.

161: refs missing

Reference cited.

164: refs missing

References cited.

167: refs missing

References cited.

168: yes, but in addition also that they generally generate single DNA cuts at well-defined places.

Yes, this is a salient point, added in revision.

169: The ancillary module of what? This sentence is a bit unclear and appears to come out of nowhere.

The ancillary module of CRISPR-Cas systems, as specified in the revision. This sentence naturally concludes the section on modules of CRISPR-Cas systems which are depicted in Box 1.

194: “New variants that were identified in recent genome and metagenome analyses that were absent in the complete genome collection were added to the list of CRISPR loci (Supplementary Figures 1-4, Supplementary Dataset 4).” → isn’t this mentioned before in line 186? Or what is the difference?

No, these sentences are not redundant. The sentence starting on l. 194 describes a specific situation with new variants missing in our complete genome collection. Edited for style but no reason to delete anything.

176/177/184/197: what kind of profiles? HMM Profiles? PSI-BLAST profiles? For clarity better to be complete.

Specified.

206: Ref? By who?

Yes, “formally recognized” was not an accurate phrase. Replaced with “explicitly added to the classification” in the revision, references cited.

208: of subtypes in this class à of type I subtypes

The comment is unclear. This is not only about type I but the entire Class 1.

213-214: perhaps the authors can also mention the role of Cas7 and Cas5 for clarity.

Roles as subunits of the effector complex mentioned in revision.

223: target and cleave instead of target (i.e. nuclease activity is not required for targeting).

Corrected.

246: can be tentatively predicted à are predicted

Modified as suggested.

248: most intriguing à This is quite arbitrary.

Yes, this sentence was non-essential, deleted.

278: Another new variant à Another new CRISPR-Cas system subtype

No, this should stay. Subtype is I-E whereas I-E1, i-E2, I-E3, etc. are variants

285: three additional (prevents double use of more in the same sentence)

Changed as suggested.

315: I think it is important to indicate here that there are many distinct effector proteins.

Amended as suggested.

323:  the known cOA sensors? Or is it evident that there are no other sensors?

There are other sensors, but those are rare. CARF/SAVED domains are by far the most abundant and well-studied. Some other sensors are indicated in Figure 3B and in the new Table 1.

325: unrelated what? CARF domains? Or ring nucleases?

Ring nucleases – rephrased to make the statement unambiguous.

328: refs missing

References cited.

371: Please rephrase this sentence requires rephrasing for clarity

Rephrased.

633-636: How do the authors envision this? Currently it feels a bit like a negative note to end on, while the authors have done great work in providing a good way for CRISPR-Cas classification. I feel that, due to shuffling and divergence, there is no 'best' way to do a classification (even if we scientists like to accurately classify everything).

Considering the reviewer's comment, this sentence was deleted.

Reviewer #2 (Remarks to the Author):

The current manuscript by Makarov et al 2025 is a comprehensive compendium of prokaryotic CRISPR systems and their classifications. It is well written, has a clear purpose and addresses the stated aims. This manuscript will undoubtedly be highly cited and be of significant interest to those involved in CRISPR research, both basic and applied. It is an extremely dense and detailed review. This does create a high barrier to entry when reading that requires a significant understanding of CRISPR biology. This is not to negate its significance but may limit interest outside of the immediate field.

I have some general criticisms regarding the preparation of the manuscript in its current state. Most figures are poorly presented, which detracts from the quality and clear impact of this manuscript. There is inconsistent graph sizing, font sizing and colour selection.

We modified most of the figures to improve their appearance.

In many instances, it is difficult to pair what is being mentioned in the text to what the figures are presenting. Many of the figures are also devoted to the variation in CRISPR genomic organisation, whilst the later sections on distribution and evolution, which I found to be extremely informative and captivating, lack suitable figures (other than figure 5 which has its pluses but is still largely left wanting).

We appreciate the reviewer's highly positive assessment of the section on CRISPR-Cas evolution. We strived to illustrate the main evolutionary trends in a simple manner, and we believe Figure 5 serves this purpose.

The sheer level of variation within subtypes, in particular Cas12, is daunting. Whilst this is extremely interesting does every variant within a subtype need distinct classification, or would a generalised principle be enough for classification (i.e. box1) with the acknowledgement that variation exists within species. For example, V-F04 to V-F13, is there need for these to be subdivided into distinct variant classes. What value does this distinction add. This is not a criticism, but merely an alternative thought for the way in which these systems are classified.

We follow the established hierarchical classification system and consistently classify all published or predicted types, subtypes and variants in each iteration of the CRISPR-Cas classification. Not 'every' discovered CRISPR-Cas system meets the criteria for a variant but only those that are, first, distinct enough from others in terms of sequence conservation and/or locus organization, and second, are sufficiently diverse to establish the boundaries of variation. We believe that these criteria make the variants useful for classification and analysis of newly discovered systems. Some published systems fail to pass these criteria and then we combine them as shown in Supplementary Table 1.

• Line 100: I would remove the word obviously. Tone seems off when it is used.

Deleted as suggested.

• Figure 1 requires improvement. There is big variation in text size and colour palettes. Why are D and E so large compared to A and B

The figure was modified, taking these suggestions into account.

- Figure 2 is hard to follow. There is a lot going on with regard to the different subtypes.
- The novelties with regard to genomic organisation for each crispr subtype are highlighted in the figure, yet variations in function (method of targeting) are only mentioned in the text. It would be helpful as a reader to have the functional consequences of this genomic divergence displayed in the same figure. Maybe a simple subheading above each subfigure is all it needs.

Indeed, there is a lot of variation among the subtypes and variants. We modified the figure to include subheadings for panels C, D and E. We are not sure how to simplify this figure further. It is just not a simple figure because it illustrates complex reality.

- Font on figure three is extremely small in places

It is 5pt or larger which is generally acceptable

- Figure 4 requires significant improvement. I like the general idea behind 4C and D but why is it only used here and not for other types. 4C and D are also poorly presented.

This approach is only used for type V because it shows the greatest functional diversity. We are not sure in what sense panels C and D are poorly presented.

- The distribution of CRISPR systems in bacteria and archaea is a really interesting section. Figure 1C though could use improvement. Maybe just some gridlines to improve clarity. Also there is no scale bar for the heatmap.

The figure was modified (also in response to comments of Reviewer 1), the scale is included.

- Figure 5 is great, and has a lot of potential to be a really important part of this review. It's a shame that the text doesn't line with the figure all that well. I think some expansion of figure 5 to highlight how different subtypes/variants are being assembled would be beneficial and make the review more accessible. Figure 5 also needs a legend.

A legend has been included. Again, the idea behind this figure is to schematically illustrate the main trends in CRISPR-Cas evolution, and in that sense, we believe the text lines with the figure suitably.

- There is large number of authors associated with this manuscript. All are significant players in CRISPR biology. Whilst this does give weight and support to this system of classification within the field, is the inclusion of all authors necessary? K.S.M., S.A.S., Y.I.W., P.M., P.P. and E.V.K. did the research and wrote the article, is this not sufficient for the authorship. I think some further justification for the number of researchers should be included.

On this matter, we respectfully but firmly and definitively disagree. Indeed, to quote the reviewer, to "give weight and support to this system of classification within the field", it is crucial for the article to represent the consensus among the key researchers, and this is what this group of authors has

achieved. We believe the current language in the Authors' Contributions section reflects this and fully justified the list of authors.

Reviewer #3

Overall, this is an authoritative study from the leaders in the CRISPR field that updates the classification of CRISPR-Cas systems and refines the possible evolutionary trajectories of the various system types. The manuscript is comprehensive in scope, well organized and easy to follow. I would have the following suggestions and comments:

We appreciate the positive assessment of our work and the constructive comments that are addressed below.

1. p. 9, lines 244-247: "Given the lack of conservation of catalytic aspartate residues in Cas7 and the presence of apparently active HD nuclease domain...can be tentatively predicted to cleave RNA targets". The lack of catalytic residues in Cas7 and presence of active HD domain would imply cleavage of DNA targets. Please correct the error.

Corrected. We regret the error and appreciate the reviewer noticing it.

2. p. 11, first and second paragraph of the Type III CRISPR system section. About 97% of type III systems are predicted to be able to produce cOA or SAM-ATP signalling molecules. How does the phylogenetic analysis of Cas10 and the resulting subclassification of type III system inform/predict which cOA species (i.e. cyclic tri-, tetra- or hexa-AMP) is the predominant product and the likely signalling molecule for the respective system? It would be helpful if the authors could expand the discussion of the classification in light of the recent studies highlighting the determinants of cOA product selectivity.

This has been investigated previously: the type of messenger molecule produced by Cas10 correlates with the Cas10 phylogeny but far from completely. We included a sentence to clarify this.

3. p. 11, third paragraph of the Type III CRISPR system section. It would be helpful to expand the description of the classification of the Crf sensor-effector proteins. Given that 3 out of the 11 Crf families include ring nucleases, is there an overlap between these families and the Crn1-Crn5 families or are these evolutionarily independent?

Yes, Crn1 is CARF protein from Crn7 family, Crf1 also has a ring nuclease activity and Crf5 is a predicted ring nuclease as indicated in Figure 3B.

It would be helpful to clarify this issue. Likewise, cOA-activated transcription factors should also be mentioned, along with their placement in the Crf family system.

These transcription factors are included in Figure 3B.

4. Fig. 4B - CARF-PD(D/E)XK nucleases have also been shown to have RNase activities as well. Consider revising the figure.

Yes, we revised the figure and the legend and included this information to the Table 1.

5. p. 18, lines 545-547. What would be the criteria/rationale for classifying HRAMP as ARAMP systems as new/separate CRISPR types as opposed to being variants of the type III system?

These are described in the previous classification paper. They are not associated with CRISPR repeats and have gene organizations that seem to be distinct enough to justify new types. We emphasize this in the revised manuscript.

The authors have put major efforts in increasing the correctness and completeness of information displayed, citations, and readability/flow. The manuscript now reads a lot better than the previous version, and the author have made clear efforts in clarifying the classification of or type III accessory proteins.

We appreciate this positive assessment of our effort.

Table 1 aims at providing the much needed clarity about the type III effector classification. While this is a good start, there are various unclarities in this table. Crf7 is listed in both the cOA sensors-effectors and ring nucleases, but the old names and superfamily descriptions are not the same. It seems like the superfamily descriptions should be the same for the same proteins. Crf9 old names are also inconsistent between these two sections of the table. Several proteins in the sensor-effector portion of the table are described as having (or possibly having) ring nuclease activity, but are not included in the ring nuclease section of the table (eg. Crf1 and 9, and CalpL). Furthermore, Crf5 is described as having effector and ring nuclease function in the ring nuclease section but is not included in the sensor-effector section of the table. As is evident from this comment, the Table still needs work to provide further clarity for the classification.

We are grateful for the reviewer's close attention to the details of this table (now Supplementary table 2). We have carefully checked all the information in the table and resolved the unclarities pointed out by the reviewer.

I still feel the manuscript meanders here and there, and I feel that the flow can be improved in order to help the reader comprehend the interesting information more easily. While this might be personal preference, I have given some more directions that can improve flow:

- Cluster insights into class 1 type I, III, and IV systems (in that order) in the first section
- First explain new insights into class 1 (I, III, IV) and class 2 (II and V, VI) systems, (and HRAMP/ARAMP), only then move on to distribution, and finally to type III effector diversity and system evolution.

We appreciate the reviewer's attention to the structure of the article. Under the requirements for the Analysis articles, the manuscript was partly reorganized. The current order of presentation is not much different from that proposed by the reviewer except that the section on type III effector diversity follows the one on class 1 systems rather than coming at the end of the Results. We still believe this provides for a better structure of the narrative.

-Additional subheadings might help readers (if allowed)

There now 4 subheading under Results, and we believe this reflects the main themes of the paper.

-It might help the reader to further refer to Box 1 and figures at multiple instances (e.g. box 1 is only referred to once while explaining the different system subunit compositions), while it is relevant for many other statements made. The same is true for statements in which it would help to refer to figures. Also Figure numbering is in the wrong order in the text - this will require careful proofreading.

Box 1 is now Figure 1 which is referred to in many places in the text. We regret and apologize for the errors in the references to the figures, this has been checked and fixed.